# LuTHy: a double-readout bioluminescence-based two-hybrid technology for quantitative mapping of protein–protein interactions in mammalian cells

Philipp Trepte[1] iD, Sabrina Kruse[1], Simona Kostova[1], Sheila Hoffmann[2], Alexander Buntru[1], Anne Tempelmeier[1], Christopher Secker[1,3], Lisa Diez[1] iD, Aline Schulz[1], Konrad Klockmeier[1], Martina Zenkner[1], Sabrina Golusik[1], Kirstin Rau[1], Sigrid Schnoegl[1], Craig C Garner[2] & Erich E Wanker[1,*] iD

## Abstract

Information on protein–protein interactions (PPIs) is of critical importance for studying complex biological systems and developing therapeutic strategies. Here, we present a double-readout bioluminescence-based two-hybrid technology, termed LuTHy, which provides two quantitative scores in one experimental procedure when testing binary interactions. PPIs are first monitored in cells by quantification of bioluminescence resonance energy transfer (BRET) and, following cell lysis, are again quantitatively assessed by luminescence-based co-precipitation (LuC). The double-readout procedure detects interactions with higher sensitivity than traditional single-readout methods and is broadly applicable, for example, for detecting the effects of small molecules or disease-causing mutations on PPIs. Applying LuTHy in a focused screen, we identified 42 interactions for the presynaptic chaperone CSPα, causative to adult-onset neuronal ceroid lipofuscinosis (ANCL), a progressive neurodegenerative disease. Nearly 50% of PPIs were found to be affected when studying the effect of the disease-causing missense mutations L115R and ΔL116 in CSPα with LuTHy. Our study presents a robust, sensitive research tool with high utility for investigating the molecular mechanisms by which disease-associated mutations impair protein activity in biological systems.

**Keywords** CSPα; LuTHy; missense mutations; protein–protein interactions; quantitative

**Subject Categories** Genome-Scale & Integrative Biology; Methods & Resources; Network Biology

**Mol Syst Biol. (2018) 14: e8071**

## Introduction

Protein–protein interactions (PPIs) play an essential role in the proper functioning of living cells (Perkins *et al*, 2010; Cafarelli *et al*, 2017). They transmit information between signaling proteins, regulate enzymatic activities of proteins, and control the cellular tasks of molecular machines (Couzens *et al*, 2013; Taipale *et al*, 2014; Arumughan *et al*, 2016). Mutation-dependent perturbations of PPIs play a crucial role in the development of diseases (Wang *et al*, 2012; Sahni *et al*, 2015). The systematic identification and characterization of interactions between human proteins is therefore of critical importance to better understand complex biological processes and molecular mechanisms of pathology (Stelzl *et al*, 2005; Bushell *et al*, 2008; Markson *et al*, 2009; Rolland *et al*, 2014; Huttlin *et al*, 2017).

Several genetic methods for the identification of PPIs in mammalian cells have been described (Buntru *et al*, 2016), including mammalian protein–protein interaction trap (MAPPIT; Lievens *et al*, 2016), fluorescence resonance energy transfer (FRET; Grünberg *et al*, 2013), single molecule pull-down (SiMPull; Jain *et al*, 2011), dual luminescence-based co-precipitation (DULIP; Trepte *et al*, 2015), avidity-based extracellular interaction screen (AVEXIS; Bushell *et al*, 2008), or mammalian membrane two-hybrid (MaMTH; Petschnigg *et al*, 2014), which all use different biophysical principles and terminal readouts for PPI detection. While in FRET-based assays, the association of proteins is monitored directly through the energy transfer of tagged fluorescent reporter proteins (Vogel *et al*, 2006), in MAPPIT or MaMTH, PPIs are detected indirectly through auxiliary reactions of reporter proteins (Petschnigg *et al*, 2014; Lievens *et al*, 2016). In some methods, for example, MaMTH, interactions are detected *in vivo* under steady state conditions, enabling the identification of transient and phosphorylation-dependent PPIs (Petschnigg *et al*, 2014). In others, mammalian cells expressing tagged hybrid proteins are lysed before binary interactions are detected in crude extracts. In

---

1   Neuroproteomics, Max Delbrück Center for Molecular Medicine and Berlin Institute of Health, Berlin, Germany
2   Synaptopathy, German Center for Neurodegenerative Diseases, Berlin, Germany
3   Department of Neurology, Charité Universitätsmedizin Berlin, Berlin, Germany
    *Corresponding author. Tel: +49 30 9406 2157; E-mail: ewanker@mdc-berlin.de

---

DULIP or LUMIER (luminescence-based mammalian interactome mapping), for example, PPIs are monitored by luminescence in co-precipitations of protein complexes (Barrios-Rodiles *et al*, 2005; Trepte *et al*, 2015). In contrast to MaMTH, these methods preferably detect PPIs with high binding affinities, while weak interactions are less efficiently identified (Trepte *et al*, 2015). Due to this diversity in procedures, underlying biophysical principles, and detection readouts, it is not surprising that largely complementary results are obtained when positive PPI reference sets are analyzed systematically (Braun *et al*, 2009; Venkatesan *et al*, 2009; Chen *et al*, 2010; Lievens *et al*, 2014). To create an improved technology for more comprehensive and sensitive identification of PPIs than currently possible, it seems advisable to combine different detection principles.

Here, we present LuTHy, a bioluminescence-based two-hybrid method that enables the detection of binary PPIs with high sensitivity and specificity in mammalian cells through the combination of two readouts in one experiment. Interactions between ProteinA-mCitrine- and NanoLuc-tagged hybrid proteins are first detected *in vivo* through quantification of BRET and subsequently *ex vivo* through a luminescence-based co-precipitation (LuC). We benchmarked LuTHy against known interactions and performed a targeted screen to identify interactions of the synaptic protein CSPα, which has a role in adult-onset neuronal ceroid lipofuscinosis (ANCL) and other neurodegenerative disorders (Nosková *et al*, 2011; Sharma *et al*, 2012; Burgoyne & Morgan, 2015). The disease relevance of our findings and further applications of LuTHy for systematic PPI mapping are discussed.

## Results

### Development of LuTHy

Our initial motivation in establishing LuTHy was to increase sensitivity of PPI detection in order to generate datasets of higher confidence and density than previously possible. To achieve this, we developed a method that can detect both weak and strong as well as direct and indirect PPIs in one procedure, using two different biophysical detection principles.

Direct interactions between proteins of interest (X and Y) are detectable in cells through quantification of bioluminescence resonance energy transfer (BRET; Pfleger & Eidne, 2006), when the tagged luciferase donors and fluorescent acceptors come into close proximity (<10 nm; Dacres *et al*, 2012). We first constructed plasmids for the co-production of NanoLuc luciferase (NL) and ProteinA-mCitrine (PA-mCit) hybrid fusion proteins in mammalian cells to measure BRET (Fig 1A and Appendix Fig S1A and B). We assumed that protein complex formation between NL-X and PA-mCit-Y should subsequently be detectable in lysed cells with a bioluminescence-based co-precipitation (LuC) assay (Fig 1A). Here, the PA-tag in the hybrid protein is used for bait precipitation and the NL-tag for the detection of the interacting prey (Fig 1A).

To prove our concept, we used the high-confidence interaction BAD/BCL2L1, shown with several PPI detection assays (Braun *et al*, 2009; Trepte *et al*, 2015). Plasmids encoding the hybrid proteins PA-mCit-BAD and NL-BCL2L1 as well as the control proteins PA-mCit, NL, PA-NL, and PA-mCit-NL were generated (Fig 1B and Dataset EV1), and recombinant protein production was confirmed by SDS–PAGE and immunoblotting (Appendix Fig S1C–E).

We first assessed the interaction between BAD and BCL2L1 hybrid proteins with in-cell BRET measurements. We co-expressed PA-mCit-BAD/NL-BCL2L1, PA-mCit-BAD/NL, PA-mCit/NL-BCL2L1, and PA-mCit/NL in HEK293 cells and quantified luminescence emission at 370–480 and 520–570 nm 48 h after transfection (Appendix Fig S2). Emission was also quantified with the positive control PA-mCit-NL and the control PA-NL, used to correct for donor luminescence bleed-through (Pfleger *et al*, 2006). BRET ratios were calculated (Appendix Fig S1F) and found to be significantly higher in cells co-producing PA-mCit-BAD/NL-BCL2L1 compared to the control interactions (Fig 1C), indicating that the BAD/BCL2L1 interaction can be detected through quantification of BRET. A high BRET ratio was also obtained for PA-mCit-NL (Fig 1C), where NL was directly linked to PA-mCit.

Next, we investigated PPI detection in LuC assays. We lysed the transiently transfected cells directly after BRET measurement and quantified fluorescence emission at 530 nm (mCit$_{IN}$) as well as the total luminescence emission after substrate addition (NL$_{IN}$) in crude protein extracts (Appendix Fig S3A), confirming the production of PA-mCit- and NL-tagged proteins. We next precipitated the PA-tagged proteins (PA-NL, PA-mCit, PA-mCit-BAD, and PA-mCit-NL) from the lysates using IgG-coated 384-well microtiter plates. After extensive washing, fluorescence (mCit$_{OUT}$) and luminescence (NL$_{OUT}$) emission was quantified in the precipitated complexes (Appendix Fig S3B). The luminescence signals obtained from crude extracts and precipitates (Appendix Fig S3A and B) were used to calculate normalized LuC ratios (Appendix Fig S1G) for all tested binary interactions. LuC ratios are indicative of the co-precipitation efficacy of NL-tagged prey proteins, which depends on expression levels and the successful precipitation of PA-tagged baits. We found a significantly higher LuC ratio for the interaction PA-mCit-BAD/NL-BCL2L1 than for the controls (Fig 1D), confirming that it is specifically detected with the LuC readout. High LuC ratios were also obtained for the hybrids PA-NL and PA-mCit-NL.

### LuTHy detects binary interactions with high specificity and sensitivity

To evaluate the performance of LuTHy more comprehensively, we examined the previously established reference sets hPRS (*Homo sapiens* positive) and hRRS (*H. sapiens* random; Braun *et al*, 2009), containing 81 and 80 PPIs, respectively (Appendix Fig S4A and B). Two expression plasmids were constructed for each protein to produce both NL- and PA-mCit-tagged fusions, allowing analysis of all proteins both as baits and preys.

In addition to potential interactions (NL-X/PA-mCit-Y), two control interactions (NL-X/PA-mCit and NL/PA-mCit-Y) were tested systematically, enabling calculation of corrected BRET (cBRET) and LuC (cLuC) ratios (Appendix Fig S1F and G) for the entire reference sets (Appendix Figs S4C and D, and S5A and B; Datasets EV2 and EV3). We obtained cBRET and cLuC ratios for 303 binary interactions from hPRS and hRRS, detecting PPIs with high reproducibility in independent experiments with both readouts (Fig 2A and B), indicating that LuTHy is suitable for larger scale screening. Receiver operating characteristic (ROC) analysis was performed to define the optimal cutoff for recovering positive interactions from hPRS

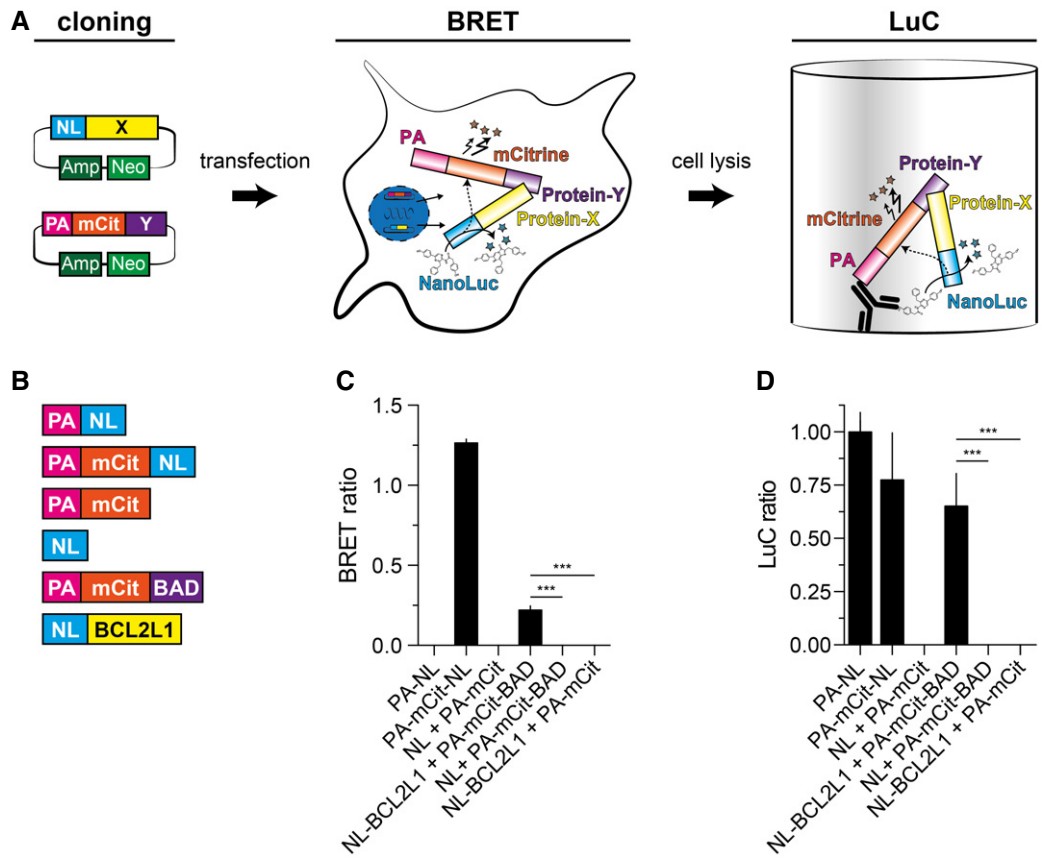

**Figure 1. The LuTHy procedure.**

A   Schematic representation of the workflow of the LuTHy method. Expression vectors encoding NL and PA-mCit-tagged hybrid proteins are cloned and co-transfected into mammalian cells. Binary interactions are detected with a double readout; first, *in vivo* with *in-cell* bioluminescence resonance energy transfer (BRET) and second, *ex vivo* with a luminescence-based co-precipitation (LuC).

B   Schematic representation of control proteins and fusion proteins of interest for investigating the interaction between PA-mCit-BAD and NL-BCL2L1 in proof-of-principle LuTHy experiments.

C   Calculated BRET ratios for the indicated protein pairs. The positive control protein PA-mCit-NL and the interacting fusion proteins NL-BCL2L1 and PA-mCit-BAD show high BRET ratios.

D   Calculated LuC ratios for the indicated protein pairs. The positive control proteins PA-NL and PA-mCit-NL and the interacting proteins NL-BCL2L1 and PA-mCit-BAD show high LuC ratios.

Data information: Data are representative of more than three independent experiments. All values are mean ± s.d. Significance was calculated by one-way ANOVA followed by Dunnett's multiple comparisons *post hoc* test; ***$P < 0.001$.

(Fig 2C). "True" interactions in this PPI set were detected with cutoffs of $\geq 0.01$ and $\geq 0.03$ for the cBRET and cLuC ratios, respectively. Next, we also assessed whether the subcellular localization of the tested interaction pairs influences the success rate of PPI detection with LuTHy assays. Using the protein pairs in hPRS and hRRS, we performed ROC analysis for known cytoplasmic, nuclear, and membrane proteins to define the optimal cutoffs for PPI detection. We found that the initially defined cutoffs of $\geq 0.01$ and $\geq 0.03$ for cBRET and cLuC ratios, respectively, are well suited to define "true" PPIs where at least one of the tested proteins per interaction pair is a cytoplasmic, a nuclear, or a membrane protein (Appendix Fig S5C–E). However, for protein pairs where both tested proteins are known to be membrane-associated or contain membrane-spanning domains more stringent cutoffs of $\geq 0.03$ (cBRET) and $\geq 0.05$ (cLuC) are required to define "true" interactions (Appendix Fig S5F). Thus, using two cutoffs, 40 binary interactions

(49.4%) were finally recovered from hPRS, of which 36 (44.4%) were identified with BRET, 22 (27.2%) with LuC and 18 (22.5%) with both readouts (Fig 2D). When combining the results from both readouts, LuTHy detects PPIs in hPRS with higher success rates than KISS (Lievens *et al*, 2014), LUMIER (Barrios-Rodiles *et al*, 2005; Braun *et al*, 2009), Y2H (Venkatesan *et al*, 2009; Chen *et al*, 2010), and several other methods (Appendix Fig S6A and C), which have success rates between 25 and 35% (Braun *et al*, 2009; Lievens *et al*, 2014). Compared to the older methods, LuTHy detected both new and previously identified PPIs in hPRS, confirming that our approach to combine two independent readouts (BRET and LuC) is a valid strategy. Using the defined cutoffs, two interactions (2.5%) were recovered from hRRS, none of which was double-positive (0.0%), substantiating that the double-readout method detects PPIs in reference sets with very high specificity and sensitivity (Appendix Fig S6B and D).

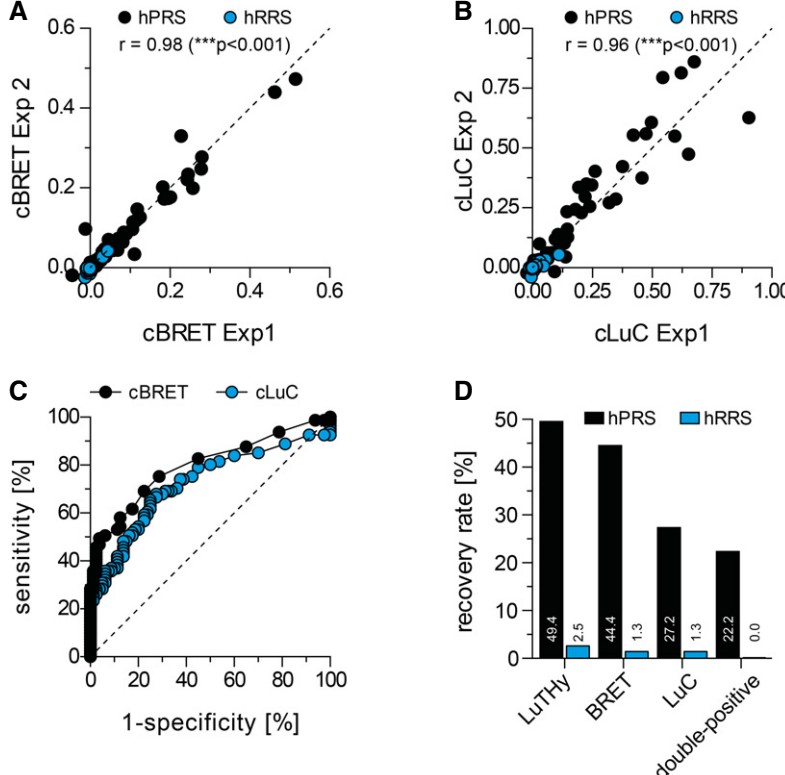

**Figure 2. Evaluation of assay performance with positive and negative binary reference sets.**

A, B    Reproducibility of LuTHy PPI mapping experiments with positive and negative reference sets. The scatter plots show the mean cBRET (A) and cLuC (B) ratios from two independent experiments (Exp 1 and Exp 2), and the calculated two-tailed Pearson correlation coefficient is indicated; ***$P < 0.001$.

C    Receiver operating characteristic (ROC) analysis of cBRET (AUC = 0.80 ± 0.04) and cLuC (AUC = 0.72 ± 0.04) data determines the thresholds to define true positive binary interactions.

D    Performance of BRET and LuC assays in benchmarking studies with hPRS and hRRS.

## LuTHy detects low- and high-affinity interactions

To investigate whether LuTHy is able to detect interactions with different binding affinities ($K_{DS}$), we next screened an affinity-based interaction reference set, termed AIRS, including 71 interactions with known $K_D$ values (Fig 3A and B and Dataset EV4). We assessed each protein in AIRS both as bait and prey in two independent experiments. Using the established cutoffs (Appendix Fig S5C–F), we found that both BRET and LuC assays in LuTHy detected AIRS PPIs with high reproducibility (Appendix Fig S7A and B). LuTHy recovered 35 (49.3%) of 71 binary interactions (Appendix Fig S7C), a similar success rate as with the hPRS (Fig 2D). We identified 31 (43.7%) interactions with the BRET readout, 19 (26.8%) with LuC (Appendix Fig S7C–E), and 15 (21.1%) with both, confirming the two components of LuTHy to yield complementary but also overlapping results.

To assess the influence of interaction strength on the recovery of PPIs, groups of interactions with low, medium, and high binding affinities (Fig 3B) were analyzed. We found that in-cell BRET assays detect strong and weak interactions with similar success rates (Fig 3C), indicating binding-strength independence. LuC, in strong contrast, recovered high-affinity interactions with significantly higher success rates than low-affinity interactions (Fig 3C and D), as

previously observed for co-precipitation-based methods (Trepte et al, 2015).

One of the high-affinity interactions ($K_D$ = 0.3 fM; Johnson et al, 2007) from AIRS, between the ribonuclease RNASE1 and its inhibitor RNH1, is readily detectable with LuC but not with BRET (Appendix Fig S8A–D). This supports binding affinity as the critical parameter for detecting PPIs with LuC. With in-cell BRET, however, resonance-energy-transfer efficiency critically depends on the proximity and the orientation of donor and acceptor molecules (Dacres et al, 2012). As FRET (Vogel et al, 2006) and DULIP (Trepte et al, 2015) assays are based on similar detection principles as the BRET and LuC assays, respectively, we used them for the validation of this specific interaction. While no significant FRET efficiency could be measured (Appendix Fig S8E), we obtained high background-corrected normalized luminescence-based interaction ratios (cNIRs) using the DULIP assay (Appendix Fig S8F–H).

## BRET50 ratios provide relative binding strengths for binary interactions

Previous studies indicate that BRET assays can determine the relative binding strength between proteins when performed as donor saturation assays, in which increasing acceptor/donor ratios are

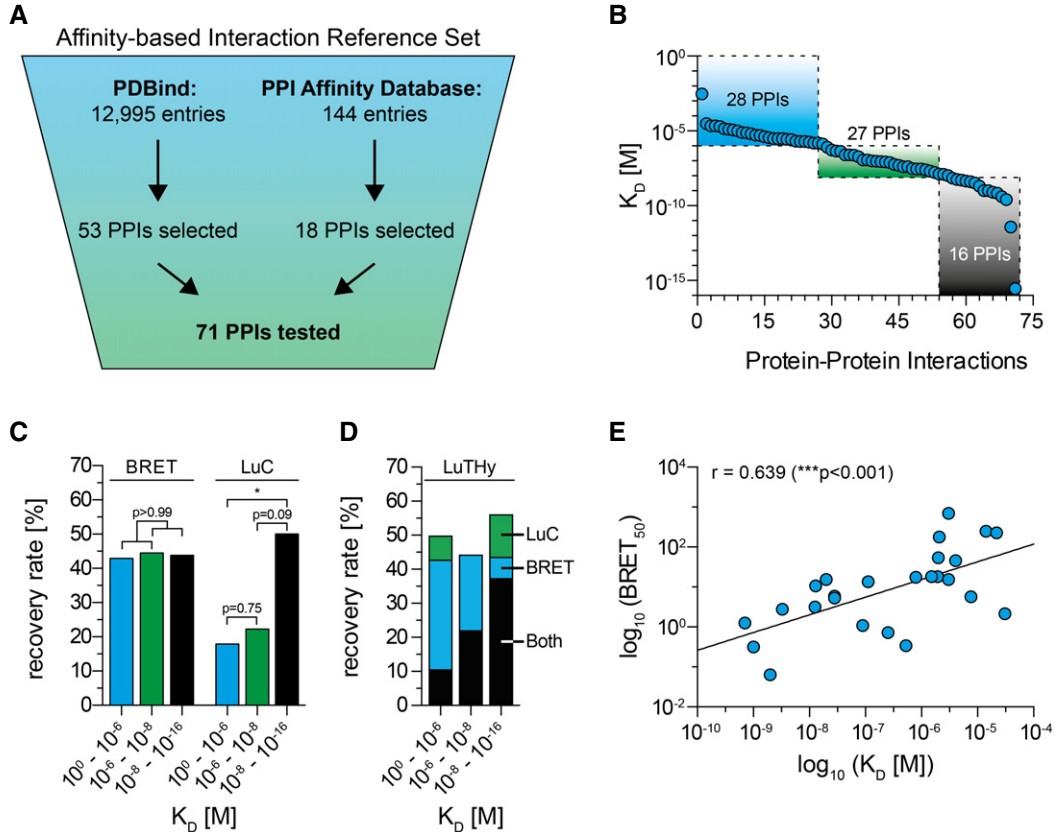

**Figure 3.  Systematic analysis of interacting proteins with known binding affinities.**

A   Selection of PPIs for the affinity-based interaction reference set (AIRS) from the PDBbind (Wang *et al*, 2004) and the Protein–Protein-Interaction Affinity Database 2.0 (Kastritis *et al*, 2011).

B   Selected protein pairs cover a broad range of published binding affinities; PPIs with low ($> 10^{-6}$ M), medium ($> 10^{-8}$ M), and high binding affinities ($\leq 10^{-8}$ M) were sub-grouped in the AIRS.

C   Recovery rates of PPIs from affinity-based sub-groups with BRET and LuC assays. Significance was calculated by two-sided Fisher's exact test; *$P < 0.05$.

D   Success of PPI detection from AIRS sub-groups considering single and double LuTHy readouts.

E   Scatter plot depicting the relationships between published binding affinities ($K_D$) and in-cell $BRET_{50}$ measurements for 25 interactions in AIRS; $BRET_{50}$ values are the mean from two independent experiments. Significance was calculated by two-tailed Spearman correlation; ***$P < 0.001$.

---

systematically analyzed (Appendix Fig S9A; Drinovec *et al*, 2012; Lavoie *et al*, 2013). We therefore produced saturation curves and calculated $BRET_{50}$ values with high reproducibility (Appendix Fig S9B) for 25 of the 31 BRET-positive interactions in AIRS (Appendix Fig S9C). Next, we plotted these values against published $K_D$ values determined with cell-free biochemical methods (Wang *et al*, 2004; Kastritis *et al*, 2011) and observed a highly significant correlation between $BRET_{50}$ and published $K_D$ values (Fig 3E). This shows that $BRET_{50}$ values are indeed indicative of protein binding affinities and can provide valuable additional information about the strengths of NL- and PA-mCit-tagged hybrid-protein interactions.

**PPIs detected with LuTHy can be confirmed by fluorescence and bioluminescence imaging (BLI)**

In order to assess whether PPIs detected with LuTHy can be validated with imaging techniques, we examined the well-studied, functional interaction between the synaptic proteins Syntaxin-1 and Munc18 (Rizo & Südhof, 2012), including the Munc18 variants with

single and double point mutations (K46E, E59K, and K46E/E59K). K46E and E59K significantly reduce the binding affinity of Munc18 to Syntaxin-1; the double mutation even prevents the interaction (Han *et al*, 2011). First, NL- and PA-mCit-tagged fusion protein expression was confirmed by SDS–PAGE and immunoblotting (Appendix Fig S10A). Systematic analysis with LuTHy revealed a strong BRET signal for the PA-mCit-Syntaxin-1/NL-Munc18-WT interaction, while lower BRET ratios were detected with NL-Munc18-K46E and -E59K (Appendix Fig S10B). No interaction was observed with the double mutant NL-Munc18-K46E/E59K, confirming published results (Han *et al*, 2011). Donor saturation curves and $BRET_{50}$ analysis confirmed the impact of the point mutations on PPI detection (Appendix Fig S10C and D).

Finally, we assessed the influence of the mutations in Munc18 on the subcellular localization of Syntaxin-1 with fluorescence microscopy and BLI (Mezzanotte *et al*, 2017). Munc18 is critical for targeting Syntaxin-1 to the plasma membrane in mammalian cells (Han *et al*, 2011). We co-expressed PA-mCit-Syntaxin-1 with NL-Munc18-WT, -K46E, -E59K, or -K46E/E59K in HEK293 cells and

analyzed the Syntaxin-1 localization with fluorescence microscopy. PA-mCit and PA-mCit-NL were investigated as controls. They were distributed throughout the cytoplasm (Appendix Fig S10E), while PA-mCit-Syntaxin-1 was predominantly located in the perinuclear region in the absence of NL-Munc18-WT (Fig 4A). In its presence, PA-mCit-Syntaxin-1 was targeted to the plasma membrane, confirming published reports (Han *et al*, 2011). Next, the NanoLuc substrate was added and cells were assessed by BLI, with luminescence emission recorded at 460 and 535 nm using appropriate bandpass (BP) filters (Fig 4A and Appendix Fig S10E). Finally, BRET images were calculated by dividing the intensities recorded at the longer wavelength by those recorded at the shorter one. We observed a strong BRET signal at the plasma membrane in cells co-producing PA-mCit-Syntaxin-1 and NL-Munc18-WT, indicating a direct interaction (Fig 4A). In comparison, BRET signals at the plasma membrane were reduced or undetectable with the mutant variants, confirming the initial BRET measurements in microtiter plates (Appendix Fig S10B and C). Interestingly, in cells co-producing NL-Munc18-K46E or -E59K, an accumulation of PA-mCit-Syntaxin-1 in the perinuclear region was observed, probably attributable to the loss of interaction at the plasma membrane.

## Effects of chemical compounds and heat stress on binary interactions

Targeting disease-relevant interactions with small molecules has great potential for the development of new medicines, but discovering drugs that potently and specifically stimulate or disrupt PPIs remains a challenge (Scott *et al*, 2016). To assess whether LuTHy can detect direct compound effects on PPIs, we first performed proof-of-principle experiments with the well-known immunosuppressant rapamycin, which was previously shown to stimulate the association of FKBP12 and the FRB domain of mTOR (Machleidt *et al*, 2015). We incubated HEK293 cells co-producing NL-FKBP12 and PA-mCit-FRB with different concentrations of rapamycin and quantified BRET and LuC ratios after 4 h. We observed a nanomolar, concentration-dependent stimulation of the interaction with BRET and LuC (Fig 4B), confirming earlier reports (Machleidt *et al*, 2015).

Next, we investigated the direct effect of Nutlin-3, an anti-cancer agent previously shown to disrupt the association between MDM2 and the tumor suppressor p53 (Lievens *et al*, 2014), on the NL-MDM2 and PA-mCit-p53 interaction. We incubated HEK293 cells with different concentrations of Nutlin-3 and monitored the association 6 h after compound addition. We observed that the compound potently inhibited ($IC_{50}$ ~8 and 11 nM) the interaction in mammalian cells (Fig 4C), confirming previously published results (Lievens *et al*, 2014).

We hypothesized that besides direct also indirect compound effects on PPIs should be detectable with the LuTHy technology. To address this question, we next investigated the effects of the small molecules ganetespib and geldanamycin on the self-association of heat shock factor 1 (HSF1) in mammalian cells. This protein is known to trimerize under heat stress conditions, leading to transcriptional activation of heat shock genes, such as *HSP70* and *HSP40* (Morimoto, 1993; Gil *et al*, 2017). Furthermore, previous studies have demonstrated that a cellular response similar to a heat shock can be induced upon treatment of cells with the Hsp90 inhibitors ganetespib and geldanamycin (Stebbins *et al*, 1997; Lin *et al*, 2008), suggesting that these compounds might indirectly influence HSF1 oligomerization. To address this, we co-expressed NL-HSF1 and PA-mCit-HSF1 in HEK293 cells and, 24 h post-transfection, treated the cells with the chemical compounds ganetespib and geldanamycin. BRET ratios were quantified after 1, 3, 6, 12, 18, 24, and 48 h in compound-treated and untreated cells. In comparison with untreated cells, we observed a significant increase in the BRET signal after 3 h in compound-treated cells, which slowly decreased over time (Fig 4D), indicating that both compounds indirectly stimulate the self-association of HSF1.

Finally, we assessed whether heat shock conditions also stimulate the association of HSF1 in mammalian cells (Rabindran *et al*, 1993). We exposed HEK293 cells co-expressing the proteins NL-HSF1 and PA-mCit-HSF1 to heat shock (42°C) and monitored the association of the fusion proteins by quantification of BRET after 90, 180, or 360 min. In addition, BRET was quantified in cells before the heat shock and after a recovery phase of 18 h at 37°C. We observed a rapid increase of the BRET signal upon heat stimulation, which decreased again to normal levels during the recovery phase at 37°C (Fig 4E). Thus, the impact of heat stress on PPIs can be readily monitored using the LuTHy method.

---

**Figure 4.   Effects of missense mutations and small molecules on PPI detection with LuTHy.**

A   Live-cell bioluminescence imaging (BLI) of HEK293 cells co-producing the indicated fusion proteins. mCitrine was excited at 500 nm, and emitted fluorescence was detected at 535 nm. After substrate addition, short (460) and long (535) band-pass (BP) filters in a dual-view adapter were used to detect the emitted luminescence simultaneously at the respective wavelengths. BRET images were calculated by dividing the 535 BP by the 460 BP images using ImageJ. Scale bar = 20 μm.

B   Quantification of the rapamycin-induced interaction between NL-FKBP12 and PA-mCit-FRB. Plasmids encoding the fusion proteins were co-transfected in HEK293 cells and 24 h later treated with the indicated concentrations of rapamycin. After an additional incubation for 4 h, BRET was quantified upon substrate addition. After cell lysis in buffer with indicated rapamycin concentrations, LuC ratios were determined. $EC_{50}$ values were obtained with nonlinear curve fitting. Data are representative of at least two independent experiments. All values are mean ± s.d.

C   Effects of Nutlin-3 on the interaction between NL-MDM2 and PA-mCit-p53. Same procedure as in (B) except for 6 h treatment with Nutlin-3. Data are representative of at least two independent experiments. All values are mean ± s.d.

D   Time resolved HSF1 oligomerization by Hsp90 inhibition. HEK293 cells co-producing NL-HSF1 and PA-mCit-HSF1 for 24 h were transferred to 384-well plates containing ganetespib or geldanamycin to reach a final concentration of 1 μM. Luminescence was measured at the indicated time points and the calculated BRET ratios normalized to the respective untreated control. Data are the mean of two biological replicates. All values are mean ± s.e.m. Significance was calculated by two-way ANOVA followed by Dunnett's multiple comparison's *post hoc* test. *$P < 0.05$; **$P < 0.005$; ***$P < 0.001$.

E   Schematic depiction of a heat shock experiment. HEK293 cells co-producing NL-HSF1/PA-mCit-HSF1 for 48 h were subjected to a heat shock at 42°C for 6 h, after which the cells were recovered at 37°C for 24 h. Luminescence was measured at the indicated time points: immediately before the heat shock, during the heat shock at 90, 180, and 360 min, and after a recovery phase of 24 h. Calculated cBRET ratios were normalized to a control plate that was not heat-treated. All values are mean ± s.e.m. from three independent experiments. Significance was calculated by one-way ANOVA followed by Dunnett's multiple comparisons *post hoc* test; *$P < 0.05$.

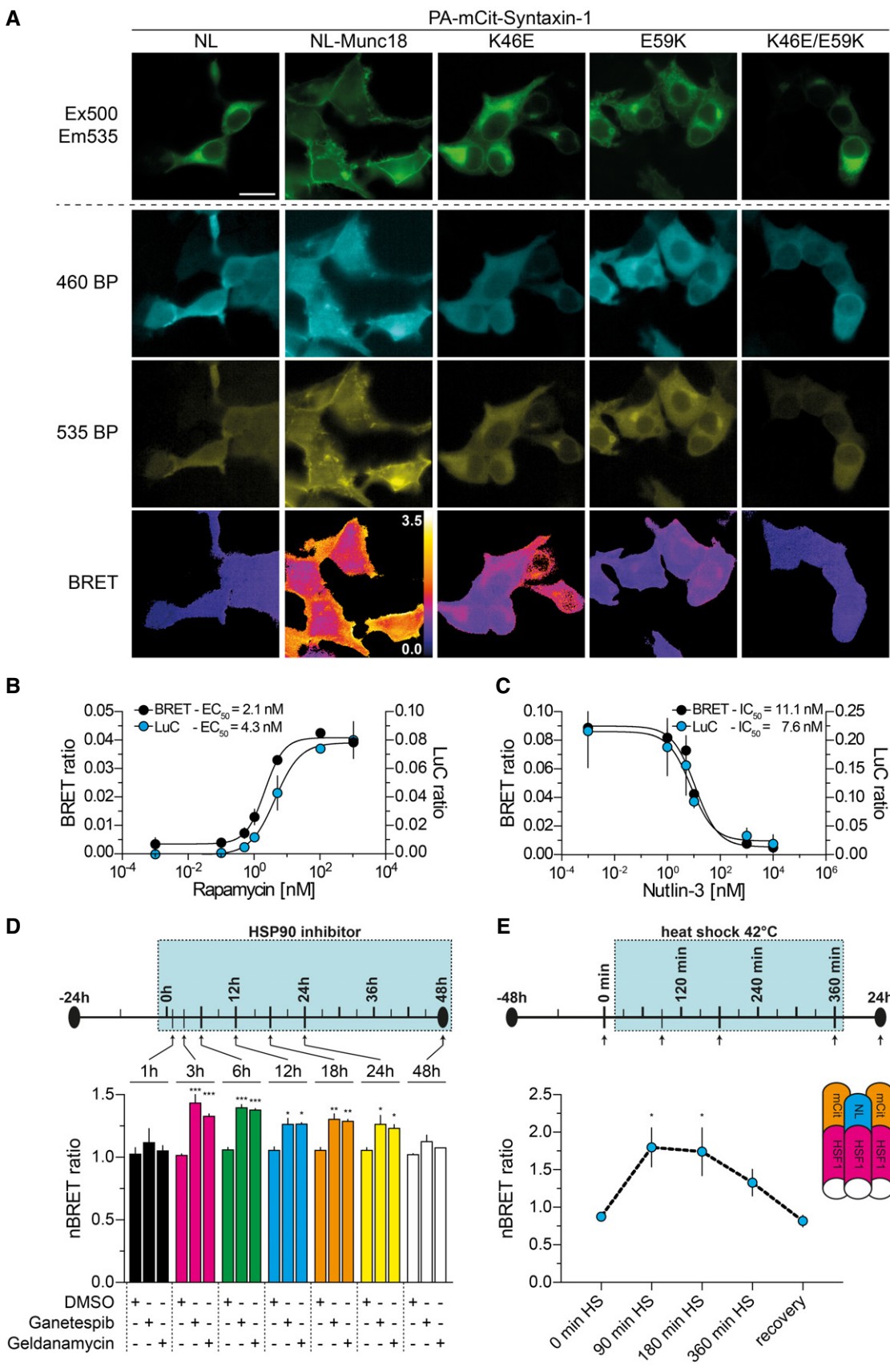

**Figure 4.**

## The positioning of fusion tags in hybrid proteins influences the success rate of PPI detection

Previous studies indicate that the positioning of fusion tags either at the N- or C-terminus of proteins of interest influences the success of PPI detection with two-hybrid methods (Stellberger *et al*, 2010). To address this question systematically, we generated expression plasmids to produce the protein VCP (valosin-containing protein) and 10 ubiquitin regulator X (UBX) domain-containing proteins (p37,

p47, UBXD1-6, UBXD8, and UBXD9) as N- and C-terminally tagged NL- and PA-mCit-fusions in mammalian cells (Fig 5A, and Datasets EV1 and EV5). Previously, interactions between VCP and all of these 10 UBX-domain-containing proteins have been reported (Raman *et al*, 2015), suggesting that they should also be detectable with the LuTHy technology. We systematically tested each interaction (e.g., between VCP and p37) in the eight possible hybrid-protein combinations, determining cBRET and cLuC ratios for 80 PPIs in HEK293 cells. We found that all reported PPIs are detectable with both BRET

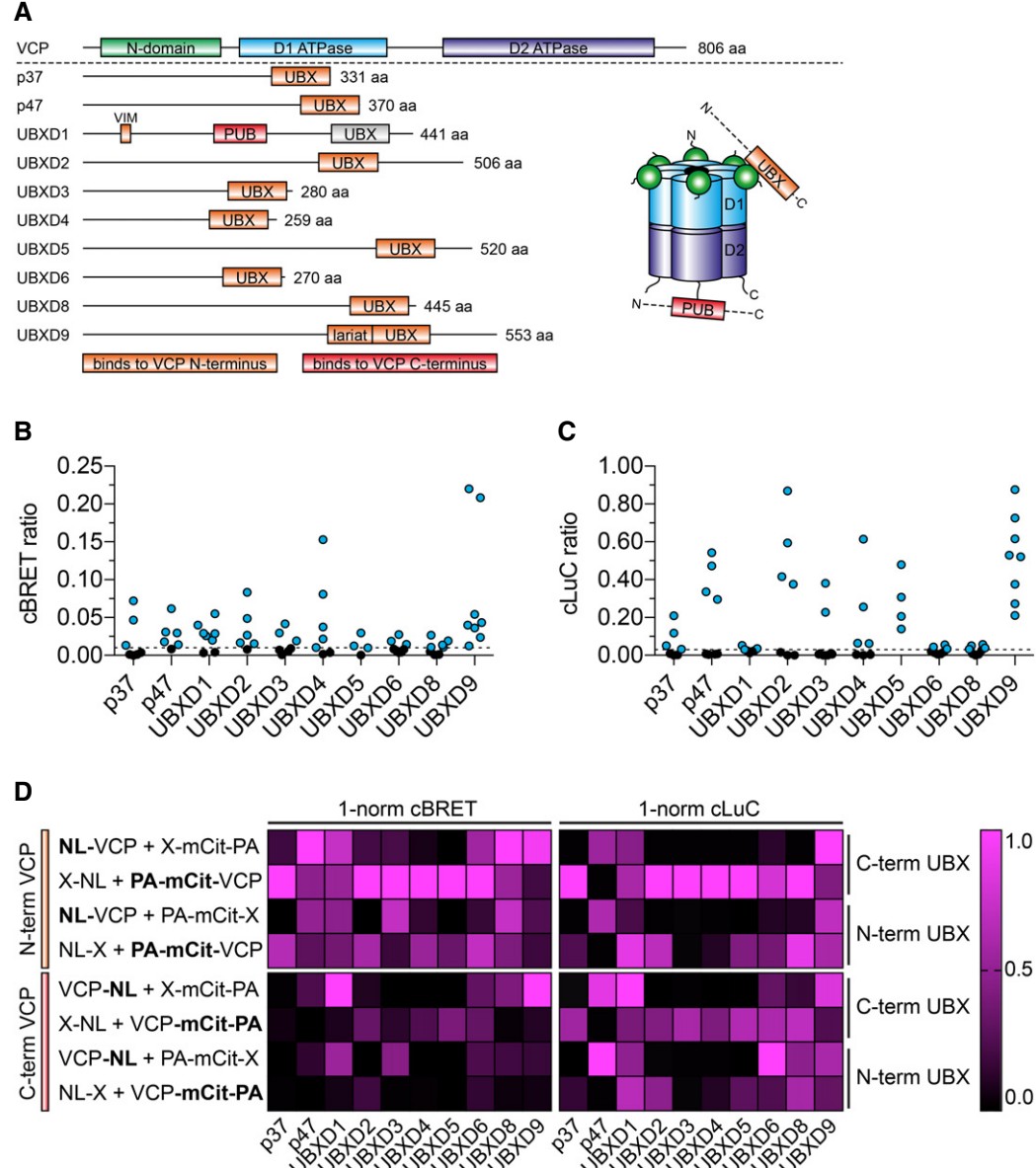

**Figure 5.  Systematic investigation of the impact of fusion-tag orientation.**

A    Domain overview of VCP and 10 UBX-domain-containing proteins. The UBX proteins bind with their C-terminally located UBX-domains to the N-domain in VCP, with the exception of UBXD1 that binds to the C-terminus of VCP via a PUB domain.

B, C   VCP and the UBX-domain-containing proteins were co-expressed as N- and C-terminal NL and PA-mCit fusion proteins in eight different orientations for which cBRET (B) and cLuC (C) ratios were generated. cBRET and cLuC values over the threshold of ≥ 0.01 and ≥ 0.03, respectively, are colored in cyan.

D    The cBRET and cLuC ratios were normalized to the highest of the obtained eight values to easily identify the orientation among the eight tested interactions with the highest cBRET and cLuC value.

and LuC measurements in LuTHy assays (Fig 5B and C), confirming the previous results (Raman *et al*, 2015). However, it is important to note that the success of PPI detection with both readouts is influenced by the site of tagging (Dataset EV5). We found that 46 (57.5%) and 41 (51.3%) of the 80 tested binary interactions scored positive in BRET and LuC assays, respectively, indicating that multiple combinations of hybrid proteins reveal positive PPIs in systematic LuTHy experiments.

It is known that UBX proteins bind with their C-terminal UBX-domain to the N-domain of VCP (Fig 5A). Accordingly, the highest cBRET ratios were obtained when VCP was tagged at its N-terminus and the UBX proteins at their C-terminus, respectively (Fig 5D). Exceptions were UBXD1, which binds to the C-terminus of VCP and UBXD9, which is known to remodel VCP into a heterotetrameric structure (Arumughan *et al*, 2016). This suggests that for BRET experiments, fusion tags should be added in close proximity to the interaction domain. In contrast, the LuC results showed no clear preference for fusion-tag orientation. However, it seems that the highest cLuC ratios are obtained when VCP was used as the bait (Fig 5D), indicating that it is crucial for PPI detection with LuC, which of the two proteins of interest contains the PA-tag.

Finally, we investigated whether the LuTHy technology also detects PPIs when the hybrid proteins are expressed in cells at similar-to-endogenous protein levels. To address this question, we focused on the interaction between VCP and UBXD9 that was repeatedly detected in BRET and LuC assays (Fig 5B–D). We first performed transfection experiments with different concentrations of plasmid DNA and determined that concentrations of 1 and 25 ng of the plasmids pcDNA3.1-NL-VCP and -UBXD9-mCit, respectively, are sufficient to detect luminescence and fluorescence activities in HEK293 cells (Appendix Fig S11A and B). Analysis of cell extracts by SDS–PAGE and immunoblotting revealed that under these experimental conditions, the recombinant proteins are expressed at endogenous protein levels or even below (Appendix Fig S11C and D). Next, we co-transfected the plasmids into HEK293 cells (1 and 25 ng) and assessed their association by quantification of BRET. In comparison with control interactions (NL-VCP/mCit or UBXD9-mCit/NL), we detected a significant BRET signal in NL-VCP/UBXD9-mCit co-expressing cells (Appendix Fig S11E), indicating that the LuTHy technology detects PPIs, even when the hybrid proteins are expressed at very low levels in cells.

## Identification of CSPα-interaction partners

To assess whether LuTHy can be applied for systematic protein interaction screening, we generated a target library of expression plasmids encoding 125 NL-tagged presynaptic proteins that play a functional role in synaptic transmission. We selected various components of the exocytosis/endocytosis machinery as well as regulators of this process (Appendix Fig S12A and Dataset EV6). In addition, we created a plasmid for the expression of PA-mCit-tagged CSPα, a molecular chaperone in the synapse (Sharma *et al*, 2011). CSPα maintains protein homeostasis in the presynapse and likely associates with multiple proteins that play a role in synaptic transmission (Zhang *et al*, 2012; Donnelier & Braun, 2014). To this day, however, only a few direct binding partners such as SNAP25 or dynamin-1 have been identified (Sharma *et al*, 2011; Zhang *et al*,

2012), suggesting that additional PPIs can be found by targeted interaction screening.

Using the established cutoffs, we systematically screened 125 protein pairs in two independent experiments (Appendix Fig S12B and C), identifying a total of 42 interactions with CSPα-mCit-PA (Fig 6A, Appendix Fig S13A and B and Dataset EV6); 36 PPIs were identified with the BRET and seven with the LuC readout (Appendix Fig S13A and B and Dataset EV6). Three (7.1%) of the 42 identified PPIs were already known (Dataset EV7), while over 90% of PPIs detected with LuTHy have not been reported. A large fraction of the detected interactors are SNARE complex members and proteins involved in vesicle fusion processes (Dataset EV8), confirming a functional role of CSPα in synaptic vesicle exocytosis (Sharma *et al*, 2011).

Our analysis revealed that a relatively large fraction (22 of 42, 52%) of the identified CSPα interaction partners are membrane-associated or membrane domain-containing proteins (Appendix Fig S14A, and Datasets EV6 and EV8). This is not unexpected due to the fact that CSPα in cells gets palmitoylated (Greaves *et al*, 2012) and probably is attached to membranes, when it interacts with NL-tagged partner proteins. To assess whether CSPα interacts specifically with membrane proteins, we performed an additional focused PPI screen with the protein PICK1 (Thomas *et al*, 2013), which, similar to CSPα, is palmitoylated in cells and attaches to membranes (Appendix Fig S14B). Quantification of cBRET and cLuC ratios confirmed the interactions between CSPα-mCit-PA and the 22 tested NL-tagged membrane-associated or membrane domain-containing proteins (Appendix Fig S14C and D, Dataset EV9). In contrast, no or very weak interactions were detected with PICK1, substantiating our result that CSPα interacts specifically with membrane proteins.

Finally, we investigated whether the interactions between CSPα and membrane-associated or membrane domain-containing proteins can be also detected in primary hippocampal neurons under endogenous conditions. We selected five positive and one negative interaction from the PPI dataset and performed *in situ* proximity ligation assays (PLAs) using target-specific antibodies. With this approach, all LuTHy positive PPIs such as the association of CSPα with SLC17A7, SYP, or SYT1 were confirmed in neuronal cells, while the LuTHy negative interaction with STX4 could not be detected (Appendix Fig S14E). This indicates that biologically relevant interactions with membrane proteins are identified with the LuTHy technology.

## CSPα preferentially associates with synaptic proteins

The relatively high success rate of PPI detection (42 of 125, 34%) in our initial interaction screen suggests that CSPα preferentially interacts with synaptic proteins involved in endo-/exocytotic processes (Dataset EV6). As confirmation, we performed an additional interaction screen with CSPα-mCit-PA and a target library of 131 mostly non-synaptic NL-tagged proteins. With this protein set, we identified a total of 18 PPIs (14%), of which 12 interactions were identified with the BRET and eight with the LuC readout (Appendix Fig S14F, Dataset EV10). Interestingly, the success rate of PPI detection with CSPα was significantly lower among non-synaptic than among synaptic proteins (Appendix Fig S14G), supporting our hypothesis that CSPα has a preference of interacting with synaptic proteins. This is in agreement with previous observations that CSPα is a

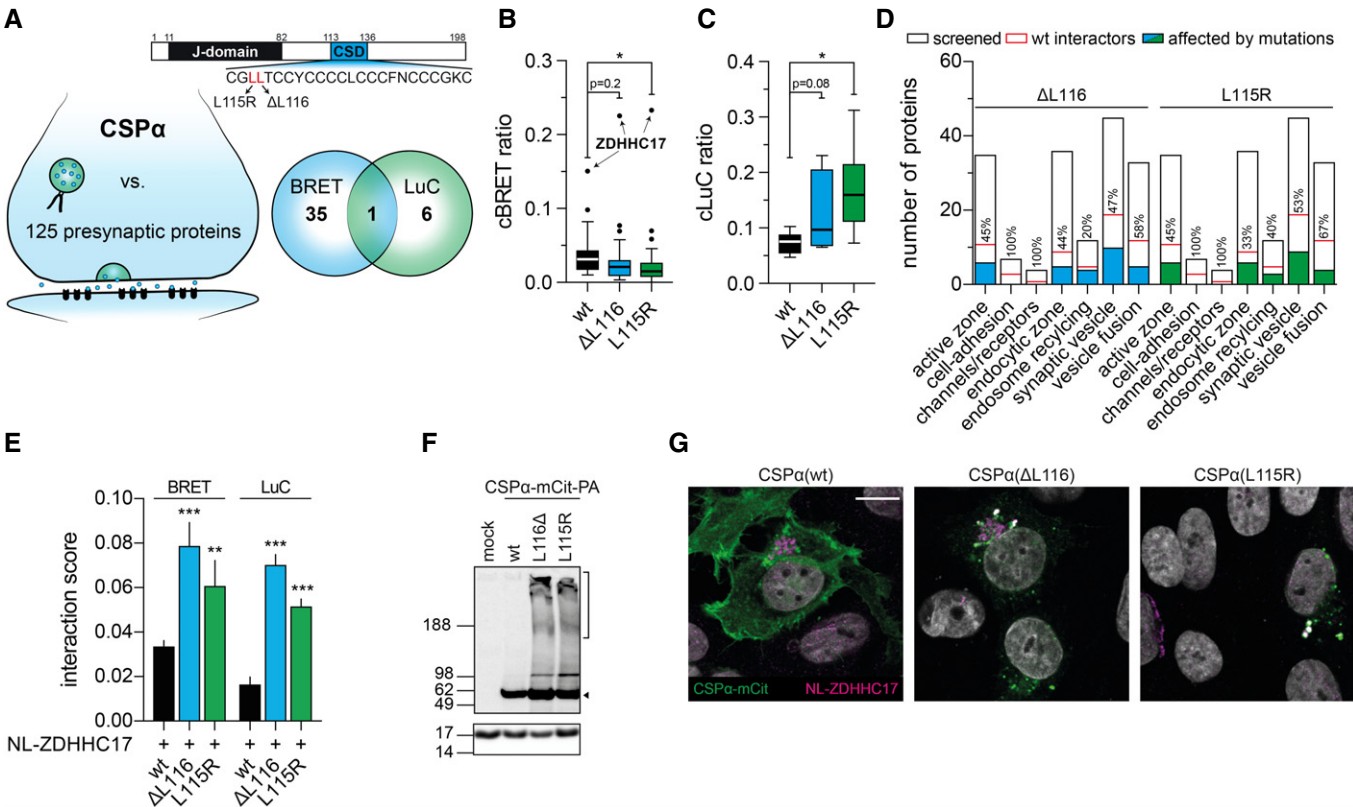

**Figure 6. Disease-causing mutations influence the association of CSPα with partner proteins.**

A Overview of the targeted PPI screen. CSPα-mCit-PA was screened in two independent experiments against a focused library of 125 NL-tagged presynaptic proteins. Venn diagram depicting the 42 identified CSPα interaction partners. Domain structure of CSPα. The J-domain and the cysteine-string domain (CSD) are depicted. The ANCL-causing mutations ΔL116 and L115R are indicated.

B, C Tukey box plots of the mean cBRET (B) and cLuC ratios (C) obtained from two independent PPI experiments with wild-type CSPα-mCit-PA and its mutant variants. Significance was calculated by one-way ANOVA followed by Dunnett's multiple comparisons *post hoc* test. *$P < 0.05$.

D Number of screened presynaptic proteins manually annotated according to their function and localization in the presynaptic terminal. Number of proteins identified with wild-type (wt) CSPα-mCit-PA or its mutant variants (ΔL116 and L115R) are indicated.

E Influence of mutations in CSPα (ΔL116, L115R) on the interactions with ZDHHC17 in primary hippocampal neurons. BRET and LuC ratios were systematically quantified for interactions with wild-type and mutant CSPα fusion proteins. Data are representative of three independent experiments. All values are mean ± s.d. Significance was calculated by two-way ANOVA followed by Dunnett's multiple comparisons *post hoc* test; **$P < 0.005$; ***$P < 0.001$.

F Immunoblot analysis of SHEP cells transfected with plasmids encoding wild-type CSPα-mCit-PA and mutant variants. Blot was developed with an anti-CSP antibody; loading control: anti-Histone-H3 antibody. Arrowhead and bracket indicate monomeric CSPα-mCit-PA and SDS-insoluble protein aggregates, respectively.

G Immunofluorescence image of SHEP cells co-transfected with NL-ZDHHC17 and CSPα-mCit-wt, -ΔL116 or -L115R constructs lacking the PA-tag. NL-ZDHHC17 was stained with an anti-NanoLuc antibody. Scale bar: 20 μm.

presynaptic co-chaperone that is critical for synapse maintenance (Sharma *et al*, 2011).

Finally, our analysis revealed that six of the 18 (33%) identified CSPα-interacting proteins in the non-synaptic protein set are membrane-associated or contain membrane-spanning domains (Appendix Fig S14H), suggesting that the chaperone has a bias for such proteins. However, statistical analysis revealed that CSPα has no significant preference for interacting with membrane or membrane-associated proteins in both tested protein stes (synaptic and non-synaptic; Appendix Fig S14H).

## Disease-causing mutations influence the association of CSPα with partner proteins

Two mutations in CSPα (ΔL116 and L115R, Fig 6A) cause adult-onset neuronal ceroid lipofuscinosis (ANCL), a rare neurological disease characterized by dementia and impairment of muscle coordination (Josephson *et al*, 2001; Nosková *et al*, 2011). To address whether these mutations influence the association of CSPα with partner proteins, we tested whether the mutant fusion proteins CSPα(ΔL116)-mCit-PA and CSPα(L115R)-mCit-PA show altered interactions with NL-tagged partner proteins in comparison with wild-type CSPα-mCit-PA. Interactions with the mutant variants in comparison with wild-type CSPα showed many altered cBRET and cLuC ratios (Fig 6B and C, Appendix Fig S15A and B and Dataset EV11), indicating that the mutations influence CSPα PPIs. In general, more interactions with membrane proteins were affected than with non-membrane proteins (Appendix Fig S15C) and in most cases the cBRET or the cLuC signals for membrane proteins were decreased (11 of 13, 85%). In comparison, many non-membrane proteins showed enhanced binding to the mutant variants of CSPα (5 of 7, 71%; Appendix Fig S15A–C and Dataset EV11).

To validate the impact of the mutations on PPIs exemplarily, we performed donor saturation assays with synaptotagmin-1 (SYT1), a calcium sensor for neurotransmitter release (Südhof & Rizo, 2011). Significantly higher $BRET_{50}$ values were obtained with the mutant variants of CSPα compared to the wild-type control protein (Appendix Fig S15D and E), confirming weaker binding to SYT1. In addition to SYT1, many proteins involved in vesicle fusion and neurotransmitter release (LIN7A, SCAMP1, STX3, SYP, SYT5, VAMP2-4) showed altered binding to mutant CSPα, suggesting that the disease-causing mutations might influence synaptic transmission in brains of ANCL patients. Using LuTHy, we found that ~50% of all interactions detected with wild-type CSPα-mCit-PA are disturbed by the two disease mutations (Fig 6D and Dataset EV11).

### The disease-causing mutations ΔL116 and L115R promote the association between CSPα and the palmitoyltransferase ZDHHC17

Using LuTHy, a strong interaction between CSPα and the palmitoyltransferase ZDHHC17 was detected with both BRET and LuC readouts, confirming previously published results (Greaves *et al*, 2012). Palmitoylation of CSPα in presynaptic terminals was reported to be critical for normal protein function (Lopez-Ortega *et al*, 2017). It occurs in a cysteine-string domain (Fig 6A), which is altered in its amino acid sequence by the disease-causing mutations ΔL116 and L115R, suggesting they might influence palmitoylation efficiency and/or the interaction with the palmitoylating enzymes. We found that the disease-causing mutations promote the association between CSPα and ZDHHC17 (Appendix Fig S15A and B), a result that was also confirmed in independent experiments by applying LuTHy in primary hippocampal neurons showing significantly increased cBRET and cLuC ratios with the mutant variants compared to wild-type CSPα (Fig 6E).

SDS–PAGE and immunoblotting of the tagged mutant CSPα fusions revealed that SDS-stable, high-molecular-weight species are detectable in cell lysates besides soluble protein migrating at ~63 kDa (Fig 6F). Such structures were not observed in cells expressing wild-type CSPα-mCit-PA, indicating that at least a fraction of the mutant fusion proteins is abnormally aggregated in mammalian cells. The formation of protein aggregates in the perinuclear region was confirmed by immunofluorescence microscopy (Fig 6G). Furthermore, we observed co-localization of ZDHHC17 with CSPα ΔL116 and L115R aggregates in cells, suggesting that the stronger interactions observed with mutant CSPα proteins in BRET and LuC assays are caused by protein co-aggregation. This view is supported by recent observations that disease-associated aggregates can recruit a variety of cellular proteins and promote their co-assembly into higher molecular weight structures (Kim *et al*, 2016).

## Discussion

LuTHy is a new double-readout bioluminescence-based two-hybrid method for systematic protein interaction mapping. Performing in-cell BRET, followed by a cell lysis step and an *ex vivo* luminescence co-precipitation assay with crude protein extract (Fig 1A), it was possible for the first time to obtain two quantitative measurements for tested protein pairs in one procedure. PPIs detected with two readouts provide more information, are potentially of higher confidence, and require less further validation (Braun *et al*, 2009) than PPIs identified with only one readout. In contrast to earlier methods (Trepte *et al*, 2015), LuTHy, with its BRET component, identifies low-affinity interactions before cell lysis, circumventing disruption of weak interactions that only occur in the intact cellular environment. In this way, LuTHy increases sensitivity and coverage considerably. Benchmarking the method with well-defined reference sets (Braun *et al*, 2009; Venkatesan *et al*, 2009), we found similar reproducibility and specificity than with previously reported single-readout methods. With a sensitivity of ~50% (Fig 2D); however, LuTHy outperforms single-readout methods like MAPPIT or LUMIER that only detect ~35% of PPIs (Braun *et al*, 2009).

Positive BRET suggests direct binding, as interactors need to come into close proximity (Pfleger & Eidne, 2006). LuC, like all precipitation approaches, does not exclude the possibility that bridging proteins mediate complex formation (De Las Rivas & Fontanillo, 2010) and exhibits a preference for high-affinity PPIs (Fig 3C). Interacting proteins detected with both readouts are likely to associate directly and strongly. However, it is important to note that potential false-positive LuTHy results could emerge from crowding effects due to high hybrid-protein expression or when both tagged proteins of interest are targeted to membrane surfaces. To circumvent this, we generally use low plasmid DNA concentrations to produce recombinant proteins in cells (Appendix Fig S11) and have also implemented a more stringent cutoff specifically for the investigation of membrane protein interactions (Appendix Fig S5F).

Quantitative PPI maps have recently been created using LUMIER with BACON (Taipale *et al*, 2012), revealing functional chaperone complexes and cellular proteostasis pathways (Taipale *et al*, 2014). Through its double readout, LuTHy should strongly advance hierarchical clustering and improve the predictions of functional protein modules (Braun *et al*, 2009).

For validation and also upscaling, we performed targeted PPI screens with libraries of > 120 synaptic or non-synaptic control proteins against the presynaptic chaperone CSPα (Donnelier & Braun, 2014). Identification of 42 synaptic proteins, with > 90% not yet described as CSPα partners, indicates that only few of the CSPα-interacting proteins in the presynapse are currently known and confirms the protein's functional role in various synaptic processes (Südhof & Rizo, 2011).

LuTHy is also highly powerful to detect stimulating and disrupting effects of chemical compounds or external stimuli on interactions. It may be a valuable tool to address this challenging area in drug discovery. Arrays of binary interactions, for example, for specific signaling cascades like MAPK (Bandyopadhyay *et al*, 2010) or ErbB (Petschnigg *et al*, 2014), could be used as quantitative sensors for systematic drug profiling with LuTHy.

Likewise, LuTHy facilitates improved characterization of disease-associated mutations, providing reliable data on increased as well as decreased binding of proteins when mutated. The impact of missense mutations has been systematically assessed with Y2H or LUMIER, with ~60% found to perturb PPIs and ~30% to cause complete loss of interactions (Sahni *et al*, 2015). Quantitative assessment with LuTHy for the disease mutations in CSPα confirmed this observation (Fig 6D).

Besides its potential to increase coverage and quality, LuTHy can be used in the future to study the dynamics of interactions induced

by specific stimuli in mammalian cells. Finally, it has an application in investigating the impact of posttranslational modifications on PPIs and signaling cascades.

In summary, LuTHy is a powerful method for improved, quantitative interactomics studies. Calculation of dimensionless BRET/LuC ratios improves comparability between labs, a challenge for all binary PPI detection methods to date. LuTHy detects PPIs without a bias for or against specific protein classes and is transferrable to a variety of cell lines. Greater depth of information per interaction in shorter time and amenability to automation will make large-scale screening campaigns considerably more time- and cost-efficient.

# Materials and Methods

**Reagents and Tools table**

| Reagent/resource | Reference or source | Identifier or catalog number |
|---|---|---|
| **Experimental models** | | |
| HEK-293 cells (*H. sapiens*) | ATCC | CRL-1573 |
| SHEP cells (*H. sapiens*) | Muth *et al* (2010) (Cancer Research) | Prof. Dr. Manfred Schwab, DKFZ, Germany |
| C57BL6/J (*M. musculus*) | Charles River/JAXTM | N/A |
| C57BL6/J (*M. musculus*) | FEM Charité | N/A |
| **Recombinant DNA** | | |
| pmTq2-DEVD-mCit | Grünberg *et al* (2013) | N/A |
| FR-20-Che~Sp1 | Grünberg *et al* (2013) | N/A |
| FK-20-Cit(K)-WW | Grünberg *et al* (2013) | N/A |
| pPA-RL-GW | Trepte *et al* (2015) | N/A |
| pGW-RL-PA | Trepte *et al* (2015) | N/A |
| pGW-FL-V5 | Trepte *et al* (2015) | N/A |
| pNL1.1 | Promega | #N1001 |
| pcDNA3.1(+) | ThermoFisher | V79020 |
| FUGW | Addgene | #14883 |
| pDONR221 | ThermoFisher | 12536017 |
| pCMV-SPORT6-DNAJC5 | Open Biosystems | EHS1001-737176 |
| pDONR223-SYT1 | The CCSB Human Orfeome Collection | 12023 |
| pDONR223-RNASE1 | The CCSB Human Orfeome Collection | 5532 |
| pDONR223-RNH1 | The CCSB Human Orfeome Collection | 3255 |
| pDONR221-mCherry | Trepte *et al* (2015) | N/A |
| pDONR221-Munc18 wt | Trepte *et al* (2015) | N/A |
| pDONR221-Munc18 K46E | Trepte *et al* (2015) | N/A |
| pDONR221-Munc18 E59K | Trepte *et al* (2015) | N/A |
| pDONR221-Munc18 K46E/E59K | Trepte *et al* (2015) | N/A |
| pDS_X-mCherry | ATCC | MBA-303 |
| pDONR221-NL | This study | N/A |
| pDONR221-FRB | This study | N/A |
| pDONR221-FKBP12 | This study | N/A |
| pDONR221-DNAJC5 | This study | N/A |
| pDONR221-SYT1ΔTM | This study | N/A |
| pDONR221-VCP | This study | N/A |
| pPA-mCit-mChe | This study | N/A |
| Additional plasmids and more information | This study | Dataset EV1 |
| **Antibodies** | | |
| Sheep gamma globulin | Jackson ImmunoResearch | 013-000-002 |

 

**Reagents and Tools table** (continued)

| Reagent/resource | Reference or source | Identifier or catalog number |
|---|---|---|
| Rabbit anti-sheep IgGs | Jackson ImmunoResearch | 313-005-003 |
| Goat anti-rabbit HRP | Sigma | A0545 |
| Goat anti-mouse HRP | Sigma | A0168 |
| Rabbit anti-NanoLuc | Promega | N/A |
| Rabbit anti-GFP | Abcam | ab290 |
| Rabbit anti-CSPα | Synaptic Systems | 154 003 |
| Rabbit anti-H3 | Abcam | ab1791 |
| Mouse anti-VCP | Progen | 65278 |
| Rabbit anti-UBXD9 | LifeSpan Biosciences | LS-C156546/EPR8616 |
| Mouse anti-tubulin | Sigma | T6074 |
| Mouse anti-VGLUT1 | Synaptic Systems | 135 311 |
| Mouse anti-VGAT | Synaptic Systems | 131 011 |
| Mouse anti-STX4 | Synaptic Systems | 110 041 |
| Mouse anti-SYP | Sigma | SVP-38 |
| Mouse anti-SYT1 | SantaCruz | sc-135574 |
| Mouse anti-VAMP2 | Synaptic Systems | 104 211 |
| Mouse IgG control | ThermoFisher | 02-6502 |
| **Oligonucleotides and sequence-based reagents** | | |
| PCR primers | This study | Dataset EV12 |
| **Chemicals, enzymes and other reagents** | | |
| Phusion High-Fidelity DNA Polymerase | ThermoFisher | F530L |
| Gateway™ LR Clonase™ II Enzyme mix | ThermoFisher | 11791100 |
| DMEM low glucose | ThermoFisher | 31885049 |
| DMEM high glucose | ThermoFisher | 41965062 |
| DMEM high glucose, HEPES, no phenol red | ThermoFisher | 21063029 |
| Fetal bovine serum (FBS) | ThermoFisher | 10270106 |
| Linear polyethylenimine (PEI), 25 kDa | Polysciences | 23966-2 |
| Cell culture microplate, 96-well white | Greiner | 655983 |
| High binding microplate, 384-well white, small volume | Greiner | 784074 |
| Coelenterazin-h | NanoLight | 301 |
| Coelenterazin-h | pjk | 102182 |
| DPBS | ThermoFisher | 14190169 |
| NanoGlo | Promega | N1120 |
| Nutlin-3 | Sigma | N6287 |
| Rapamycin | SantaCruz | 3504A |
| FuGENE® HD | Promega | E2311 |
| Geldanamycin | Selleckchem | S2713 |
| Ganetespib | Selleckchem | S1159 |
| Dual-Glo® | Promega | E2980 |
| NuPAGE™ NOVEX™ 4-12% Bis-Tris | ThermoFisher | NP0323BOX |
| Nitrocellulose membrane | GE Healthcare | 10401197 |
| WesternBright Quantum | Advansta | 12042-D20 |
| ibidi® μ-Slide 8-well | ibidi | 80827 |
| X-tremeGENE 9 | Sigma-Aldrich | 000000006365779001 |
| Duolink® *In Situ* Detection Reagents Orange | Sigma-Aldrich | DUO92007 |

**Reagents and Tools table** (continued)

| Reagent/resource | Reference or source | Identifier or catalog number |
|---|---|---|
| GeneJet™ | SignaGen | SL100488 |
| Pierce™ BCA Protein assay | ThermoFisher | 23225 |
| **Software** | | |
| GraphPad Prism 7.0c | https://www.graphpad.com | |
| SerialCloner 2-6-1 | http://serialbasics.free.fr/Serial_Cloner.html | |
| ImageJ | https://imagej.nih.gov/ij/index.html | |
| **Other** | | |
| Tecan Infinite M200 | Tecan | |
| Tecan Infinite M1000 | Tecan | |
| Tecan Infinite M1000Pro | Tecan | |
| Tecan Infinite Spark | Tecan | |
| Fujifilm LAS-3000 | Fujufilm | |
| Olympus IX83 with Andor Zyla camera and Dual-View™ adapter | Olympus, Andor and Optical Insights | |

## Methods and Protocols

### Plasmid construction

Destination vectors encoding N- and C-terminally tagged fusion proteins were generated based on the pcDNA3.1 (+) vector (Invitrogen). For the acceptor vectors pcDNA3.1-PA-mCit-GW, pcDNA3.1-GW-mCit-PA, and pcDNA3.1-PA-mCit, the coding sequences for mCitrine (mCit) and ProteinA (PA) were amplified from pmTq2-DEVD-mCit (kindly provided by Dr. Raik Grünberg, Center for Genomic Regulation, Barcelona) and pPA-RL-GW (Trepte *et al*, 2015), respectively. For N-terminal fusions, cDNAs encoding PA and mCit were amplified using the primers #1, to #4, respectively. For generation of plasmids encoding C-terminal fusions, cDNA fragments encoding PA and mCit were amplified using the primers #5 and #6 as well as #7 and #8, respectively. For the control vector pPA-mCit, which contains a stop codon after the mCit coding sequence, the primers #1 and #2 were used to amplify the PA-tag and the primers #3 and #9 to amplify the mCit tag. For the donor vectors pcDNA3.1-NL-GW, pcDNA3.1-GW-NL, and pcDNA3.1-NL, the NanoLuc coding sequence was amplified from pNL1.1 (Promega). For plasmids encoding N-terminal and C-terminal fusions, cDNAs were amplified using the primers #10 and #11 as well as #12 and #13, respectively. For the control vector pNL, the primers #10 and #14 were used to amplify a fragment encoding cmyc-NanoLuc. For all destination vectors, the resulting PCR fragments were cloned into pcDNA3.1 (+) (Invitrogen) via NheI/EcoRI/HindIII restriction sites. For control vectors and N- or C-terminal fusion vectors, the restriction sites EcoRI/XhoI/HindIII were used. Subsequently, the Gateway cassette was amplified from pGW-RL-PA (Trepte *et al*, 2015) to generate the N-terminal fusion vectors pcDNA3.1-PA-mCit-GW and pcDNA3.1-NL-GW using the primers #15 and #16. Similarly, for the creation of C-terminal fusions vectors pcDNA3.1-GW-mCit-PA and pcDNA3.1-GW-NL the primers #17 and #18 were utilized. The resulting PCR fragments encoding the Gateway cassette were cloned in the generated vectors containing the PA-mCit or NL coding sequences. For the N-terminal fusion vectors, the restriction sites HindIII/XhoI, and for the C-terminal fusion vectors, the restriction sites NheI/HindIII were used. To generate the vector pcDNA3.1-PA-NL, the cDNA encoding the PA-tag was amplified using the primers #19 and #20; the cDNA fragment encoding NanoLuc was amplified using the primers #21 and #22. The resulting PCR products were cloned simultaneously in pcDNA3.1 (+) (Invitrogen) via the restriction sites NheI/EcoRI/HindIII. The vector pmCherryGW (Invitrogen) was digested using the restriction sites XhoI and SgrAI. Then, it was ligated with the self-annealed primer #23 to obtain the plasmid pmCherry lacking the Gateway cassette. Lentiviral LuTHy plasmids were generated by the Viral Core Facility (Charité, Universitätsmedizin Berlin, Germany) based on the FUGW plasmid (FUGW was a gift from David Baltimore, Addgene plasmid #14883). The respective reporter genes and Gateway cassettes were cloned from the pcDNA3.1-based LuTHy vectors to generate pLenti PA-mCit-GW, pLenti-GW-mCit-PA, pLenti-NL-GW, and pLenti-GW-NL.

To generate the entry plasmids pDONR221-NL, pDONR221-FRB, pDONR221-FKBP12, pDONR221-DNAJC5 (CSPα) pDONR221-SYT1ΔTM, and pDONR221-VCP the sequences encoding the proteins of interest were PCR amplified and the resulting fragments were shuttled into pDONR221 (Invitrogen) using the BP clonase (Invitrogen). The cDNA encoding NanoLuc was amplified using the primers #24 and #25 from vector pNL1.1 (Promega). The FRB fragment encoding the amino acids 2,021–2,113aa of mTOR was amplified using the primers #26 and #27 from the vector FR-20-Che~Sp1. FKBP12 was amplified using the primers #28 and #29 from the vector FK-20-Cit(K)-WW, which was kindly provided by Dr. Raik Grünberg (Center for Genomic Regulation, Barcelona). The coding sequence of CSPα/DNAJC5 was amplified using the primers #30 and #31 from the pCMV-SPORT6-DNAJC5 vector (Open Biosystems, EHS1001-7373176). To create the SYT1, the entry plasmid lacking the vesicular and transmembrane domain, base pairs 241–1,266 (amino acids 81–422) of SYT1 (CCSB-12023) was amplified using the primers #32 and #33. The coding sequence of VCP lacking the stop codon was amplified from the entry clone IOH52832 using the primers #34 and #35. To generate the CSPα deletion mutation

ΔL116 and the single point mutation L115R (Nosková *et al*, 2011), the vector pDONR221-DNAJC5 was PCR amplified with the phosphorylated primers #36 and #37 (ΔL116) or #38 and #39 (L115R), respectively. The PCR products were ligated and transformed into Mach1 competent *Escherichia coli* cells. The DULIP expression plasmids and the entry vectors pDONR221-mCherrry, pDONR221-Munc18-wt, -K46E, -E59K, and -K46E/E59K were generated previously (Trepte *et al*, 2015).

For LuTHy experiments, all cDNAs were shuttled from the entry vectors into the LuTHy destination vectors (pcDNA3.1 or pLenti) using the LR clonase technology according to the manufacturer's instructions (Invitrogen). Similarly, for DULIP and FRET assays, the cDNAs from the entry vectors pDONR223-RNASE1 (CCSB-5532) and pDONR223-RNH1 (CCSB-3255) were shuttled into the destination vectors pGW-RL-PA, pGW-FL-V5, or pDS_X-mCherry (ATCC® MBA-303™). The plasmid encoding the PA-mCit-mChe fusion protein was generated by LR recombination using the pDONR221-mCherry and pPA-mCit-GW vectors. All used primers can be found in Dataset EV12.

### Cell culture, transfection, and lentivirus preparation

The human embryonic kidney cell line 293 (HEK293) was grown in low-glucose (1 g/l) DMEM (Gibco®, ThermoFisher) for DULIP and FRET experiments and in high glucose (4.5 g/l) for LuTHy assays. In both cases, media were supplemented with 10% heat inactivated fetal bovine serum (Gibco®, ThermoFisher) and cells were grown at 37°C and 5% $CO_2$. Cells were subcultured every 3–4 days and transfected with linear polyethylenimine (25 kDa, Polysciences) using the reverse transfection method according to the manufacturer's instructions. For LuTHy transfections, cells were seeded in phenol-red-free, high-glucose DMEM media (Gibco®, ThermoFisher) supplemented with 10% heat inactivated fetal bovine serum. Transfections were performed with a total DNA amount of ~200 ng per well in a 96-well plate. If expression plasmid concentration was below 200 ng/well, pcDNA3.1 (+) was used as carrier DNA to obtain the total DNA amount of 200 ng.

For LuTHy experiments in hippocampal neurons, all procedures involving animals were approved by the animal welfare committee of Charité Medical University and the Berlin state government. First, hippocampi were dissected from WT mice P0-2 brains in cold Hanks' salt solution (Millipore), followed by a 30-min incubation in enzyme solution [DMEM (Gibco, ThermoFisher Scientific), 3.3 mM cysteine, 2 mM $CaCl_2$, 1 mM EDTA, 20 U/ml papain (Worthington)] at 37°C. Papain reaction was inhibited by the incubation of hippocampi in inhibitor solution DMEM, 10% fetal calf serum (ThermoFisher Scientific), 38 mM BSA (Sigma-Aldrich), and 95 mM trypsin inhibitor (Sigma-Aldrich) for 5 min. Afterward, cells were triturated in NBA (neurobasal-A medium, 2% B27, 1% glutamax, 0.2% P/S, ThermoFisher Scientific) by gentle pipetting up and down.

All lentiviral particles were provided by the Viral Core Facility of the Charité Berlin and prepared as described previously (Lois *et al*, 2002). Briefly, HEK293T cells were co-transfected with 10 μg of shuttle vector, 5 μg of helper plasmids pCMVdR8.9, and 5 μg of pVSV.G using X-tremeGENE 9 DNA transfection reagent (Roche Diagnostic). Virus containing cell culture supernatant was collected after 72 h and filtered for purification. Aliquots were flash-frozen in liquid nitrogen and stored at −80°C.

### Creation of reference sets

The CCSB reference sets hPRS and hRRS were previously described (Braun *et al*, 2009). The previously described affinity-based reference set (AIRS) was extended by 14 PPIs (Trepte *et al*, 2015), by selecting 18 PPIs from PDBbind (Wang *et al*, 2004) and 53 PPIs from the Protein-Protein Interaction Affinity Database 2.0 (Kastritis *et al*, 2011). The cDNAs encoding 71, 81, and 80 protein pairs in AIRS, hPRS, and hRRS were shuttled into LuTHy expression plasmids for systematic interaction testing.

### LuTHy experiments

1) Reverse transfect HEK293 cells in white 96-well microtiter plates (Greiner, 655983) at a density of 4.0–4.5 × $10^4$ cells per well. Plasmids encoding donor and acceptor proteins are transfected at a 1:10 to 1:20 ratio, with 5–10 ng DNA for the donor and 100 ng for the acceptor. Primary hippocampal neurons are plated on poly-l-lysine coated white 96-well microtiter plates (Greiner, 655983) at a density of 1 × $10^4$ cells per well. Infect neurons with 5–35 μl of the viral solution (0.5–1 × $10^6$ IU/ml) per well 2–4 days post-plating and culture in NBA at 37°C and 5%, for a total of 13–15 days (days *in vitro*, DIV) before starting measurements.

2) Measure mCitrine fluorescence 48 h after transfection in intact cells (Ex/Em: 500 nm/530 nm).

3) Add coelenterazine-h (NanoLight, 301 or pjk, 102182) to a final concentration of 5 μM. Incubate cells for an additional 15 min and measure the total, short-WL and long-WL luminescence. Here, fluorescence and luminescence were measured using the Infinite® microplate readers M200, M1000, or M1000Pro (Tecan) using the BLUE1 (370–480 nm) and the GREEN1 (520–570 nm) filters at 1,000 ms integration time.

4) After luminescence measurements in intact cells, the luminescence-based co-precipitation (LuC) is performed. Lyse cells in 50–100 μl HEPES-phospo-lysis buffer (50 mM HEPES, 150 mM NaCl, 10% glycerol, 1% NP-40, 0.5% deoxcholate, 20 mM NaF, 1.5 mM $MgCl_2$, 1 mM EDTA, 1 mM DTT, 1 U Benzonase, protease inhibitor cocktail (Roche, EDTA-free), 1 mM PMSF, 25 mM glycerol-2-phosphate, 1 mM sodium orthovanadate, 2 mM sodium pyrophosphate) for 30 min at 4°C.

5) Production of PA-mCit- and NL-tagged fusion proteins is monitored by measuring fluorescence ($mCit_{IN}$) and luciferase activity ($NL_{IN}$) in crude cell lysates in white, small-volume 384-well microtiter plates (Greiner, 784074). Add 5 μl coelenterazine-h to 5 μl of cell lysates to a final concentration of 10 μM and measure the luminescence activity as before in a microplate reader with 100 ms integration time.

6) Coat small-volume 384-well microtiter plates (Greiner, 784074) the day before use with sheep gamma globulin (Jackson Immuno-Research, 013-000-002) in carbonate buffer (70 mM $NaHCO_3$, 30 mM $Na_2CO_3$, pH 9.6) for 3 h at room temperature. Block with 1% BSA in carbonate buffer before incubating with rabbit anti-sheep IgGs in carbonate buffer (Jackson ImmunoResearch, 313-005-003) overnight at 4°C. Equilibrate all wells with lysis buffer immediately before use. Do not store IgG-coated plates longer than 24 h.

7) Incubate 15 μl of cell lysate for 3 h at 4°C in the IgG-coated 384-well plates. Wash all wells three times with lysis buffer and measure the mCitrine fluorescence ($mCit_{OUT}$) as above. Finally, 15 μl of PBS buffer containing 10 μM coelenterazine-h is added to each well and the luminescence activity ($NL_{OUT}$) is measured as described above.

### LuTHy donor saturation experiments

1) For donor saturation experiments, plasmid DNAs (between 0.1 and 20.0 ng) encoding NL fusion proteins are co-transfected with increasing amounts of plasmid DNAs encoding the PA-mCit-tagged constructs.
2) BRET measurements are performed as described before, 72 h after transfection using NanoGlo® (Promega) at a final concentration of 1:500.
3) Calculate the fluorescence to luminescence ratio for the PA-mCit-NL tandem construct. In this fusion protein, the stoichiometric ratio of mCitrine and NL is 1:1. Next, the fluorescence to luminescence ratios for the studied interactions of interest can be estimated by normalizing the calculated values to the value obtained with the tandem construct.

### LuTHy data analysis

The BRET and LuC ratios from *in vivo* BRET and *ex vivo* co-precipitation measurements are calculated as follows:

1)

$$\text{BRET ratio} = \frac{\text{LWL}}{\text{SWL}} - \text{Cf},$$

with LWL and SWL being the detected luminescence at the long (520–570 nm) and the short (370–480 nm) wavelengths, respectively. Cf is the correction factor for the donor bleed-through, which is the ratio of the luminescence measured at LWL and SWL in cells expressing the PA-NL construct and calculated as:

$$\text{Cf} = \frac{\text{LWL}_{\text{PA−NL}}}{\text{SWL}_{\text{PA−NL}}}.$$

2) The corrected BRET (cBRET) ratio is calculated by subtracting either the BRET ratio of control 1 (NL/PA-mCit-Y) or of control 2 (NL-X/PA-mCit) from the BRET ratio of the studied interaction of interest (NL-X/PA-mCit-Y). The calculated BRET ratios obtained for the controls 1 and 2 are always compared with each other, and the higher value is used to correct the BRET ratio of the interaction of interest (see Appendix Fig S1F).
3) For the LuC readout, the obtained luminescence precipitation ratio (PIR) of the control protein PA-NL ($PIR_{\text{PA-NL}}$) is used for data normalization, which is calculated as follows:

$$\text{PIR}_{\text{PA−NL}} = \frac{\text{NL}_{\text{OUT}}}{3 \times \text{NL}_{\text{In}}},$$

with $NL_{OUT}$ being the total luminescence measured after co-IP and $NL_{IN}$ the luminescence measured in cell extracts directly after lysis.
4) Subsequently, LuC ratios are calculated for all interactions of interest and normalized to the $PIR_{\text{PA-NL}}$ ratio:

$$\text{LuC ratio} = \frac{\text{NL}_{\text{OUT}}/3 \times \text{NL}_{\text{IN}}}{\text{PIR}_{\text{PA−NL}}}.$$

5) Finally, a corrected LuC (cLuC) ratio is calculated by subtracting either the LuC ratio of control 1 (NL/PA-mCit-Y) or of control 2 (NL-X/PA-mCit) from the LuC ratio of the studied interaction of interest (NL-X/PA-mCit-Y). The calculated LuC ratios obtained for the controls 1 and 2 are always compared with each other and the higher value used to correct the LuC ratio of the interaction of interest (see Appendix Fig S1G).

A receiver operating characteristics (ROC) curve was obtained for PPIs of the hPRS and hRRS using corrected cBRET and cLuC ratios. Binary interactions in all LuTHy screens (hPRS, hRRS, AIRS, UBX-domain-containing, synaptic, non-synaptic) were scored positive with cBRET and cLuC ratios $\geq 0.01$ and $\geq 0.03$, respectively. Interactions between two integral-membrane or membrane-associated proteins were scored positive with cBRET and cLuC cutoffs $\geq 0.03$ and $\geq 0.05$, respectively.

Gene ontology (GO) analysis was performed using the database for annotation, visualization, and integrated discovery (DAVID) (Huang *et al*, 2009), https://david.ncifcrf.gov.

### Investigating the effects of small molecules on PPIs with LuTHy

For the investigation of the effects of small molecules on PPIs with LuTHy, HEK293 cells were co-transfected with plasmids encoding the proteins NL-MDM2 and PA-mCit-p53 or NL-FKBP12 and PA-mCit-FRB in white 96-well microtiter plates (Greiner, 655983). Increasing concentrations of Nutlin-3 (Sigma, N6287) or rapamycin (Santa Cruz, 3504A) were added to cells 24 h after transfection. Luminescence measurements in intact cells were performed after an additional incubation for 6 (Nutlin-3) or 4 h (rapamycin) upon addition of 5 μM coelenterazine-h. After the measurement of BRET in intact cells, the cells were lysed with HEPES-phospho-lysis buffer that contained the respective concentration of compounds. Finally, the compound effects on PPIs were assessed by LuC as described above.

For Hsp90 inhibition, cells were transfected in a 10-cm dish using FuGENE (Promega) according to manufacturer's instructions. Geldanamycin (Selleckchem S2713) and ganetespib (Selleckchem S1159) were added to a 384-well plate 24 h after transfection and cells seeded on top at a density of 15,000 cells per well. BRET measurements were performed in a Tecan Spark reader using filter settings at 360–485 nm and 520–650 nm at the indicated time points by coelenterazine-h injection. For heat shock experiments, HEK293 cells were transferred 48 h after transfection to 42°C for 6 h followed by a recovery phase at 37°C for an additional 24 h. Control treatment was conducted with identically transfected cells, which were constantly kept at 37°C. BRET measurements at the depicted time points (Appendix Fig S10D) were performed as described above.

### DULIP assay

The DULIP assay and data analysis were performed as described before (Trepte *et al*, 2015). Briefly, HEK293 cells were reversely transfected and lysed after 48 h. Production of PA-RL- and FL-tagged fusion proteins was monitored by measuring the respective luciferase activities in crude cell lysates. In parallel, cell lysates

were incubated for 3 h at 4°C in IgG-coated 384-well plates. Measurement of firefly and *Renilla* luminescence activities was performed using an Infinite® M1000 (Tecan) plate reader and the Dual-Glo® Stop & Glow® reagents (Promega). NIR and cNIR values were determined as described previously (Trepte *et al*, 2015).

### FRET assay

FRET assays were performed as described before (Trepte *et al*, 2015). Briefly, HEK293 cells were reversely transfected using 100 ng plasmid DNAs encoding PA-mCitrine- and mCherry-tagged fusion proteins in black 96-well microtiter plates at a density of $4.5 \times 10^4$ cells per well. 72 h after transfection, cells were fixed with 4% paraformaldehyde in PBS and washed twice with PBS. Fluorescence signals were detected with an Infinite® M1000Pro (Tecan) plate reader: donor channel [excitation (Ex)/emission (Em): 500/530 nm], acceptor channel (Ex/Em: 580/610 nm) and FRET channel (Ex/Em: 500/610 nm). In brief, FRET efficiencies ($E_{Aapp}$ in %) were calculated according to the sensitized emission formula as follows:

$$E_{Aapp} = \frac{(DA - c_D \times DD - c_A \times AA)}{AA},$$

with DD = donor channel signal; AA = acceptor channel signal; DA = FRET channel signal; $c_D$ = donor bleed-through using donor only sample (pPA-mCit); $c_A$ = acceptor cross-excitation using acceptor only sample (pmCherry).

### Western blotting

For Western blotting experiments, cells were lysed for 30 min at 4°C in HEPES-phospho-lysis buffer and the protein concentrations determined using the Pierce™ BCA assay (Thermo Scientific). Then, the same amounts of protein lysates were loaded onto NuPAGE™ Novex™ 4–12% Bis-Tris precast polyacrylamide gels (Thermo-Fisher) according to the manufacturer's instructions. Following, proteins were transferred onto a nitrocellulose membrane (GE, 10401197) using a BioRad® wet blotting system. The membrane was blocked in 3%-milk PBS-T (phosphate-buffered saline, 0.05% Tween) for 60 min at room temperature and incubated with primary antibody in 3%-milk PBS-T overnight at 4°C. Bound antibodies were detected with an appropriate HRP (horse-radish-peroxidase)-coupled secondary anti-rabbit (1:2,000, Sigma A0545) and anti-mouse (1:2,000, Sigma A0168) antibody by measuring chemiluminescence in a Fujifilm LAS-3000 after the addition of WesternBright Quantum (Advansta, 12042-D20). The primary antibodies used were as follows: anti-NanoLuc (rabbit, 1:5,000, kindly provided by Promega), anti-GFP (rabbit, 1:5,000, abcam ab290), anti-CSP (rabbit, 1:1,000, Synaptic Systems, 154 003), anti-Histone H3 (rabbit, 1:5,000, abcam ab1791), anti-VCP (mouse, 1:1,000, Progen, 65278), anti-UBXD9 (rabbit, 1:1,000, LifeSpan Biosciences, LS-C156546, EPR8616), and anti-tubulin (mouse, 1:4,000, Sigma, T6074).

### Bioluminescence imaging (BLI)

HEK293 cells were seeded at a low density with $3 \times 10^4$ cells per well in an ibidi® µ-Slide 8-well tissue culture dish and transfected using X-tremeGENE (Roche®) according to the manufacturer's instructions. Forty-eight hours after transfection, imaging was

performed using an Olympus® inverted microscope IX83 with an Andor Zyla camera and a 60× objective. A Dual-View™ adapter (Optical Insights®) between microscope und camera was equipped with a short (460/50, Chroma®) and long (535/50, Chroma®) wavelength filter. Imaging was performed in tyrode's buffer (100 mM NaCl, 5 mM KCl, 2 mM HEPES, 2 mM CaCl₂, 4 mM MgCl₂, 33 mM glucose) containing 1:50 NanoGlo® (Promega) substrate. Before substrate addition, mCitrine was excited with a LED excitation system (CoolLED® pE, 2%) using a YFP-excitation filter (Olympus®, AHF F48-003 ET-Set); the emitted fluorescence was detected with an exposure time of 100 ms. Bioluminescence imaging started immediately after substrate addition with an exposure time of 3,000 ms. Images were processed with ImageJ (http://imagej.nih.gov/ij/). Fluorescence images were background-corrected with the contrast adjusted for each image individually as the fluorescence signal was only used to study the localization of mCit-tagged proteins. The bioluminescence images, which contain the short (460 BP) and long wavelength (530 BP) signals, were split into two separate images. These two images were stacked and aligned using the StackReg plugin in ImageJ (Thévenaz, 1998). Next, the images were smoothed by median filtering, background-subtracted, and thresholded on luminescence intensities. To calculate the BRET ratio on a pixel-by-pixel basis, the 530 BP image was divided by the 460 BP image using the image calculator and presented in pseudo-color for better visualization.

### Proximity ligation assay

All animals were sacrificed according to the permit given by the Office for Health Protection and Technical Safety of the regional government of Berlin (LaGeSo, X9009/14) and in compliance with regulations laid down in the European Community Council Directive. One day before preparation of primary neurons, all coverslips containing culture plates were coated with poly-DL ornithine hydrobromide (PO, 0.005% PO in H₂O) over night at 37°C. On the preparation day, the coating solution was removed and plates were incubated with DMEM containing 10% FCS for several hours. The pregnant mice were sacrificed by cervical dislocation and the embryos were removed and placed in ice-cold standard salt solution (SSS, 135 mM NaCl, 5 mM KCl, 1.25 mM CaCl₂, 0.5 mM MgCl₂, 0.5 mM MgSO₄, 1 mM NaHCO₃, 10 mM HEPES, 25 mM glucose, 1× AB/AM). Brains of all embryos were isolated and collected in SSS on ice. Next, the meninges were removed and hippocampi of all embryos were isolated. The tissue was cut in smaller pieces and treated with trypsin solution (1× PBS-CMF, 20 mM glucose, 15 mM HEPES, 1× AB/AM, EDTA 1 mM, 3,428 BAEE/ml trypsin) for 5 min at 37°C. The reaction was stopped using DNase-Ovomucoid solution (Hanks MEM, 0.44 mM MgCl₂, 25 mM HEPES, 20 mM glucose, 50 mM MgSO₄, 30 mg/ml trypsin inhibitor, DNase 1,380 U/ml). The tissue was triturated step wise to generate a single cell suspension. The cells were counted and plated in 24-well plates containing glass coverslips at an initial density of 68.000/cm². The hippocampal cultures were maintained in B27- and 1% FCS-supplemented neurobasal medium (neurobasal medium, 25 µM β-mercaptoethanol, 0.25 mM L-glutamine, 2% B27-supplement, 0.05% penicillin/streptomycin, 1% FCS). On day *in vitro* (DIV) 4, cells were treated with AraC (5 µM) to eliminate glia cells from the culture. Neurons were fixed on cover slips at 14 DIV using 4% PFA in PBS for 15 min, followed by fixation in methanol fix (100 mM MES, pH

6.9, 1 mM EGTA, 1 mM $MgCl_2$, 90% MeOH). Following, cells were blocked in 0.1% saponin (Sigma-Aldrich), 1% BSA (VWR), 4% goat serum (Dako), 50 ng/µl RNase A (Promega), 11 ng/µl poly (A) (P9403 Sigma-Aldrich), and 2.5 ng/µl salmon sperm DNA (Invitrogen) in PBS for 30 min at room temperature. Proximity ligation assays (PLAs) were performed according to manufacturer's protocol (Duolink® *In Situ* Detection Reagents Orange, Sigma-Aldrich). Primary antibodies were prepared in antibody diluent PBS containing (0.1% saponin, 3% BSA, and 1:1,000 salmon sperm DNA) and incubated with slides (50 µl/sample) for 1 h at 37°C in a humidity chamber. Unbound primary antibodies were removed by washing the slides 3× in PBS with 0.1% saponin. Thereafter, both PLA probes (Duolink® Anti-Rabbit PLUS and Anti-Mouse MINUS) were diluted 1:5 in antibody diluent and allowed to sit for 20 min before being added to cover slips (40 µl/sample) for 1 h at 37°C in a humidity chamber. Slides were washed for 2 × 5 min in wash buffer A (0.01 M Tris, 0.15 M NaCl, 0.05% Tween) under gentle agitation to remove unbound PLA probes. For the ligation, cover slips were incubated with Duolink® ligation reagent (Ligation stock 1:5 in antibody diluent plus ligase, 1:40; 40 µl/sample) for 30 min at 37°C in a humidity chamber. Washing was performed 2 × 2 min with wash buffer A under gentle agitation. For signal amplification, Duolink® Amplification stock was diluted 1:5 in high-purity water and polymerase (1:80) was added before slides were incubated (40 µl/sample) for 100 min at 37°C in a humidity chamber. Final washing steps were performed with 2 × 10 min 1× wash buffer B (0.2 M Tris, 0.1 M NaCl) and 1 × 1 min 0.01× wash buffer B. Coverslips were mounted in fluorescent mount medium (Dako) and images acquired on a Leica TCS SP5 confocal microscope in the Advanced Light Microscopy (ALM) technology platform at the Max Delbrück Center for Molecular Medicine. Images were processed with ImageJ.

The primary antibodies used were as follows: anti-CSPα (rabbit, 1:100, Synaptic Systems, 154003), anti-VGLUT1/SLC17A7 (mouse, 1:250, Synaptic Systems, 135311), anti-VGAT/SLC32A1 (mouse, 1:200, Synaptic Systems, 131011), anti-STX4 (mouse, 1:100, Synaptic Systems, 110 041), anti-SYP (mouse, 1:100, Sigma, SVP-38), anti-SYT1 (mouse, 1:50, SantaCruz, sc-135574), anti-VAMP2 (mouse, 1:250, Synaptic Systems, 104211), and IgG isotype control (mouse, 1:2,500, ThermoFisher, 02-6502).

### Immunofluorescence imaging

SHEP cells (Muth *et al*, 2010) were seeded at low density ($3 \times 10^5$ cells) onto fibronectin (10 µg/ml)-coated coverslips in a 6-well plate and transfected using GeneJet™ (SignaGen, SL100488) according to the manufacturer's instructions. Forty-eight hours after transfection, the cells were fixed with 4% paraformaldehyde in PBS with 1:5,000 Hoechst (ThermoFisher, 33342). Cells were permeabilized with 0.1% saponin in PBS for 10 min and blocked with 3% BSA, 0.1% saponin in PBS for 30 min at RT. Primary anti-NanoLuc (rabbit, 1:1,000, Promega) and secondary Cy5-coupled anti-rabbit (1:500) antibodies were diluted in 3% BSA, 0.1% saponin in PBS and applied successively to the coverslips for 1 h at 37°C with three washing steps in between. Coverslips were mounted in fluorescent mount medium (Dako) and images acquired on a Leica TCS SP5 confocal microscope in the Advanced Light Microscopy (ALM) technology platform at the Max Delbrück Center for Molecular Medicine. Images were processed with ImageJ.

### Data availability

The protein interactions from this publication have been submitted to the IMEx (http://www.imexconsortium.org) consortium through IntAct (Orchard *et al*, 2014) and assigned the identifier IM-26441.

**Expanded View** for this article is available online.

### Acknowledgements

This work was supported by grants from the German Research Foundation (SFB740), the Federal Ministry of Education and Research (Integrament, ERA-NET NEURON), the European Union (EuroSpin and SynSys), the Berlin Institute of Health (Collaborative Research Grant) and the Helmholtz Initiative on Personalized Medicine (iMED) to E.E.W, the CHDI foundation to E.E.W and P.T. and from the German Research Foundation SFB958 and the German Center for Neurodegenerative Diseases (DZNE) to C.C.G. The relevant grant numbers are SFB740: 740/2-11, Integrament: 01ZX1314C, ERA-NET NEURON: 01W1301, EuroSpin: Health-F2-2009-241498, SynSys: HEALTH-F2-2009-242167 and BIH: 1.1.2.a.3. The funders had no role in the study design, the collection and analysis of data or the preparation of the manuscript. We thank Promega for providing an aliquot of the anti-NanoLuc antibody and Prof. Dr. Manfred Schwab (DKFZ Heidelberg, Germany) for providing the SHEP cells. We thank the Advanced Light Microscopy (ALM) technology platform of the Max Delbrück Center, where immunofluorescence imaging was performed and the Viral Core Facility of the Charité Universitätsmedizin Berlin that generated the lentiviral LuTHy vectors and particles.

### Author contributions

PT and EEW conceived the study, and PT designed all experiments and analyzed the data. SKr, AT, SKo, MZ, KR, and SG generated the expression vectors, and SKr performed most LuTHy experiments with technical assistance by MZ, AT, KR, and SG. Donor saturation experiments were performed by SKr, immunoblotting by SKr and SKo and the compound studies by SKr and SG. Heat shock studies with HSF1 were designed and performed by SKo and PT. PLA was established by LD and performed by SG. Immunofluorescence imaging was performed by AB, AT, and CS. The FRET experiment was performed by SKo. Under supervision by CCG, SH performed bioluminescence imaging studies and LuTHy experiments in hippocampal neurons. Primary neuronal cultures were prepared by AS for PLA experiments and by SH for LuTHy assays. PT, AB, and EEW conceptually designed the assay. SKr, AB, SKo, AT, and KK assisted in the design, analysis, and interpretation of experiments. All authors discussed the results and EEW, SS, and PT wrote the manuscript with comments from all authors.

### Conflict of interest

The authors declare that they have no conflict of interest.

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
