## [Review Process File · Molecular Systems Biology]

LuTHy: a double-readout bioluminescence-based two-hybrid technology for quantitative mapping of protein-protein interactions in mammalian cells

Authors: Philipp Trepte, Sabrina Kruse, Simona Kostova, Sheila Hoffmann, Alexander Buntru, Anne Tempelmeier, Christopher Secker, Lisa Diez, Aline Schulz, Konrad Klockmeier, Martina Zenkner, Sabrina Golusik, Kirstin Rau, Sigrid Schnoegl, Craig C. Garner & Erich E. Wanker

Review timeline:

Submission date:	24 th November 2017
Editorial Decision:	8 th January 2018
Revision received:	31 st May 2018
Editorial Decision:	6 th June 2018
Revision received:	8 th June 2018
Accepted:	15 th June 2018

Editor: Maria Polychronidou

Transaction Report:

1st Editorial Decision

8th January 2018

Thank you again for submitting your work to Molecular Systems Biology. We have now heard back from the three referees who agreed to evaluate your study. As you will see below, the reviewers appreciate that the presented approach is a potentially valuable contribution to the interactomics field. They raise however a series of concerns, which we would ask you to address in a revision of the manuscript.

The reviewers' recommendations are rather clear so I think that there is no need to repeat all the points listed below. As reviewer #2 mentions, some follow up analyses providing support for the biological significance of (some of) the identified CSPα interactions would significantly enhance the overall impact of the study. If you would like to discuss any of the reviewers' comments in further detail please do not hesitate to contact me.

REFeree REPORTS

Reviewer #1:

The manuscript entitled "A quantitative bioluminescence-based technology reveals the impact of disease-associated mutations on synaptic CSPα interactions" by Trepte et al, utilizes a double-readout bioluminescence-based two-hybrid technology called LuTHy, which provides two quantitative scores in one experimental procedure for testing binary protein-protein interactions in

mammalian cells. Specifically, quantification of binary PPIs is achieved using BRET and co-IP approaches. Using this double read-out, the authors show they can detect interactions with higher sensitivity than traditional single read-out methods. Furthermore, using LuTHy, the authors identified 47 interactions for the chaperone CSP α , and almost 70% of these PPIs were found to be perturbed in the disease-causing L115R substitution and L116 deletion mutants of CSP α .

Overall, the study and technology are interesting, as LuTHy can be used to detect both weak and strong interactions as well as direct (BRET) and indirect (co-IP) PPIs in one experiment, using two different biophysical detection principles. Such an approach would result in the detection of binary PPIs with high sensitivity and specificity in mammalian cells, which offsets the limitations of traditional single read-out assays to detect binary PPIs. Since much of this paper is a method development and proof-of-principle of the approach, the following are suggestions I recommend to enhance the manuscript:

Comment 1: The authors conducted a thorough analysis using different reference sets of the applicability of LuTHy. From the different PPIs analyzed, it would be interesting to see how accurate LuTHy is at determining both positive/negative PPIs in different cellular compartments. For example, is LuTHy able to detect with high confidence, PPIs that occur in the nucleus, plasma membrane, cytosol, etc? We recommend the authors try and break down the localization of the different PPIs analyzed and potentially provide ROC curves for different cellular compartments. This would be interesting as some of the currently available technologies can only detect PPIs in specific cellular compartments (ie. MaMTH is specifically used for membrane PPIs).

Comment 2: Have the authors tried performing ligand stimulation experiments to assess the effect on downstream PPIs? It would be interesting to see how LuTHy is able to detect PPIs that are induced via a receptor-ligand interaction (ie. When cells are treated with EGF, how does it affect the ability of EGFR to interact with downstream signalling proteins).

Comment 3: How amendable is LuTHy in other cell lines; specifically, ones that are difficult to transfect? Have the authors tried using a cell line other than HEK293 to show the scalability of this approach? It would be interesting to see how LuTHy can be used to detect PPIs in other cell lines.

Comment 4: One of the limitations of LuTHy is the need for the transfection of both bait and prey proteins. Have the authors tried generating stable bait or prey cells to offset the need to transfect both plasmids? Even a stable inducible bait or prey cell system would be interesting to see how the assay performs. In addition, a stable inducible cell line would also be more amendable for prospective high throughput drug screening applications.

Reviewer #2:

Summary

Trepte et al. describe a new protein/protein interaction (PPI) assay suitable for detecting and quantifying binary interactions in both live cells and in lysates. Their assay combines two well-established assays, BRET and LUMIER, into one vector system which facilitates almost simultaneous detection of interactions with two complementary methods. They benchmark the assay with a collection of known interactors (PRS) and random reference set (RRS) in addition to a panel of binary interactions with known dissociation constants.

The first part of the manuscript describes and benchmarks the method, whereas the second part focuses on the interactions of one protein, CSP α , involved in synapse function.

General remarks

There are many well-established interaction assays that scientists can use, but the problem is that there is no single assay that works in every situation. In particular, transient interactions that are difficult to detect but biologically highly relevant have been challenging to study with many existing methods. Therefore, it is important that new assays are still developed and rigorously benchmarked.

Technically, the experiments described in the manuscript are well controlled and clearly presented, and the results are (mostly) convincing. I only have a few comments about the technical aspects of the manuscript (listed below).

As a method development paper, this manuscript is almost entirely technical in nature. The authors do not perform any follow-up experiments with the interactors detected in the CSPa screen, and therefore the biological significance of the interactions remains to be determined.

Major points

The number of interactors detected in the CSPa screen is, to me, alarmingly high (38% of tested pairs!). While it is possible that this is true (given that CSPa is a chaperone), it would be important to control this more rigorously. I suggest the authors test another membrane protein or membrane-associated protein for interactions with the same set to make sure these interactions are not due to crowding at the two-dimensional surface to which membrane proteins are tethered.

In the PRS/RRS experiment, were the positive hits from both orientations included in the analysis? In all previous assays (Y2H, MAPPIT, LUMIER etc), the interactions were tested in only one orientation, so comparison to the union of two different orientations in LuTHy is not appropriate. It would be better to show the performance of the assay with both orientations and compare them separately to the other established assays. In addition, I would suggest the authors add a supplementary figure where they plot the interaction scores from one orientation against the other to get an idea of the robustness of the assay.

Because BRET is highly dependent on the proximity of the donor and acceptor, it would be nice to have an indication as to how much the interactions change when proteins are tagged in the C versus N terminus. "Where should I tag my favourite protein?" is a very common question for everyone working with fusion proteins, and such data would be very helpful to anyone who wants to use LuTHy assay in the future. I would encourage the authors to add such data if they have it.

The authors state that the CSPa preferentially targets membrane proteins, since 88% of interactors were of such kind. Please test if this is a significant enrichment over the starting pool, since synaptic proteins are likely enriched in membrane proteins to begin with.

It is not clear to me how the proteins screened for CSPa interactions were tagged (N terminus or C terminus). If they were tagged in the N terminus (as it seems), I would be very worried about proper trafficking of many membrane proteins since this would disrupt the signal sequence. Please comment.

Minor points

The authors state (page 14) that LuC is less suitable for detecting changes in interactions due to mutations. However, they only tested with LuC those interactions that were detected as changed by BRET. Thus, it is possible that 1) LuC is less suitable (as they claim), 2) BRET is less suitable because the changes it detected are not real, or 3) LuC is more suitable because it would have detected other changed interactions that BRET did not detect, or 4) they are equally suitable.

The authors state that BRET detects direct interactions, because the donor and the acceptor have to be in close proximity. This is incorrect, since they can be in close proximity even if the interaction is not direct.

On page 6 (and in several graphs), the authors mention measuring luminescence. Technically, the emitted light is not only luminescence but a combination of luminescence (photons emitted by Nanoluc) and fluorescence (photons emitted by mCitrine).

Page 10: co-produced -> co-expressed

Reviewer #3:

The LuTHy assay described by Trepte and colleagues combines two orthogonal protein-protein interaction (PPI) assays, Bioluminescence Resonance Energy Transfer (BRET) and luminescence-based co-precipitation (LuC), as a new improved method for the mapping of PPIs. This clever approach deals with a typical validation strategy for PPIs detected with a particular assay, i.e. the confirmation in an orthogonal assay. The authors rightfully point out that the high heterogeneity underlying PPIs requires complementary approaches and they refer to important papers in the field. The authors indeed suggest to combine the data from the two assays as a robust sensitive research tool to obtain a comprehensive view on the interactome.

The different aspects related to state-of-the-art binary interactomics are nicely addressed in this study: small molecule effects, weak interactions, donor saturation assays for binding constants, and interactome profiling of pathogenic variants.

The manuscript is clearly written and well structured. Figures are clear and to the point. Sufficient additional figures are provided to assess validity of all the claims. Overall quality of the work is very high.

While experimentally sound, and a nice contribution to the binary interactomics field, one important aspect is not handled in this manuscript. The plasmids used in this study contain CMV promoters. Combined with PEI-based transfections in HEK293 cells, it can be assumed that extremely high levels of both bait and prey are obtained in the cells. This can have different effects on the cells themselves (various levels of overexpression stress) and can also lead to artifacts such as false positives. While this is controlled to some extent by the random reference set (and the non-synapse set), this point should be addressed in the manuscript. Although the donor saturation assay uses increasing levels of one of the partners, it is not the best way to show this. Ideally, the authors should show the system for lower expression levels. Lentiviral transduction or stable integration of bait and prey can be considered, preferably in a different cellular background (for instance: in the SHEP cells). I have mentioned higher that an orthogonal approach is a good validation strategy. Actually most studies now indeed aim at orthogonal validation, but at the endogenous level. As an alternative, the authors can consider to evaluate some of the found interactions for CSP α at the endogenous level.

It is clear however that forced expression is a typical drawback with most binary approaches.

Minor points:

Please explain how the selection was made to obtain the panel of interactions for the affinity range.

What criteria (if any) were used?

How do the cut-offs defined in the materials and methods (p26 first paragraph; ≥ 0.03 and ≥ 0.05) relate to the data in the table EV6?

Figure 4A, Figure 5D: Is the data normalized for expression levels?

Please also address what kind of false positives can be expected for the individual approaches, and for the combined assay (if any)

1st Revision - authors' response

31st May 2018

Reviewer #1:

The manuscript entitled "A quantitative bioluminescence-based technology reveals the impact of disease-associated mutations on synaptic CSP α interactions" by Trepte et al, utilizes a double-readout bioluminescence-based two-hybrid technology called LuTHy, which provides two quantitative scores in one experimental procedure for testing binary protein-protein interactions in mammalian cells. Specifically, quantification of binary PPIs is achieved using BRET and co-IP approaches. Using this double read-out, the authors show they can detect interactions with higher sensitivity than traditional single read-out methods. Furthermore, using LuTHy, the authors identified 47 interactions for the chaperone CSP α , and almost 70% of these PPIs were found to be perturbed in the

disease-causing L115R substitution and L116 deletion mutants of CSP α .

Overall, the study and technology are interesting, as LuTHy can be used to detect both weak and strong interactions as well as direct (BRET) and indirect (co-IP) PPIs in one experiment, using two different biophysical detection principles. Such an approach would result in the detection of binary PPIs with high sensitivity and specificity in mammalian cells, which offsets the limitations of traditional single read-out assays to detect binary PPIs. Since much of this paper is a method development and proof-of-principle of the approach, the following are suggestions I recommend to enhance the manuscript:

Comment 1: The authors conducted a thorough analysis using different reference sets of the applicability of LuTHy. From the different PPIs analyzed, it would be interesting to see how accurate LuTHy is at determining both positive/negative PPIs in different cellular compartments. For example, is LuTHy able to detect with high confidence, PPIs that occur in the nucleus, plasma membrane, cytosol, etc? We recommend the authors try and break down the localization of the different PPIs analyzed and potentially provide ROC curves for different cellular compartments. This would be interesting as some of the currently available technologies can only detect PPIs in specific cellular compartments (ie. MaMTH is specifically used for membrane PPIs).

We appreciate this important suggestion by the reviewer. We have now categorized the interactions in the hPRS and hRRS reference sets using the available information about their subcellular localization (see **Appendix Tab. S2 and S3**). We now also provide ROC curves for the recovery of PPIs with cytoplasmic, nuclear and membrane proteins (see **Appendix Fig. S5C-F**). Our analysis revealed that PPIs where both proteins are known to be membrane-associated or contain membrane-spanning domains are indeed recovered from the reference sets with a lower specificity than PPIs with nuclear or cytoplasmic proteins. We have therefore adjusted the cBRET and cLuC cutoffs for the detection of such PPIs to decrease the possibility to detect false-positive interactions (see main text **page 7, line 18 to page 8, line 2**).

Comment 2: Have the authors tried performing ligand stimulation experiments to assess the effect on downstream PPIs? It would be interesting to see how LuTHy is able to detect PPIs that are induced via a receptor-ligand interaction (i.e. When cells are treated with EGF, how does it affect the ability of EGFR to interact with downstream signalling proteins).

We thank the reviewer for bringing up this important point. Since EGF ligand stimulation for the EGF receptor has been previously shown to work in BRET assays (for example Schiffer, H. H. et al. (2007) Pharmacology and signaling properties of epidermal growth factor receptor isoforms studied by bioluminescence resonance energy transfer. *Molecular Pharmacology* 71, 508–518), we investigated whether the oligomerization of the heat shock factor HSF1

can be stimulated indirectly by the treatment of cells with small molecules or heat. The results are presented now in **Fig. 4D and E** and described in the text (see **page 12, line 20 to page 13, line 19**). Our studies strongly indicate that external stimuli can indeed significantly influence the association of tagged proteins in mammalian cells.

Comment 3: How amendable is LuTHy in other cell lines; specifically, ones that are difficult to transfect? Have the authors tried using a cell line other than HEK293 to show the scalability of this approach? It would be interesting to see how LuTHy can be used to detect PPIs in other cell lines.

To answer this question, we generated new lentiviral expression vectors (**Appendix Fig. S1B**), from which we produced lentiviral particles and infected primary hippocampal neurons. To this point we have only used them to validate the effects of the disease-causing mutations in CSP α on the interaction with ZDHHC17. The new data are now presented in **Fig. 6E**. Importantly, the mutation-induced increase of cBRET and cLuC values could be validated with LuTHy in primary neurons (see **page 19 line 22 to page 20, line 2**). These results suggest that LuTHy can be applied in other cell lines besides HEK293 cells, including cells that are difficult to transfect, such as primary neurons.

Comment 4: One of the limitations of LuTHy is the need for the transfection of both bait and prey proteins. Have the authors tried generating stable bait or prey cells to offset the need to transfect both plasmids? Even a stable inducible bait or prey cell system would be interesting to see how the assay performs. In addition, a stable inducible cell line would also be more amendable for prospective high throughput drug screening applications.

To this point, we have not generated and tested stable cell lines. However, this is planned for future, larger scale PPI and drug screening studies with LuTHy. Because tagged proteins of interest can now be produced with lentiviral expression vectors, the rapid generation of stable lines is clearly feasible.

Reviewer #2:

Summary

Trepte et al. describe a new protein/protein interaction (PPI) assay suitable for detecting and quantifying binary interactions in both live cells and in lysates. Their assay combines two well-established assays, BRET and LUMIER, into one vector system which facilitates almost simultaneous detection of interactions with two complementary methods. They benchmark the assay with a collection of known interactors (PRS) and random reference set (RRS) in addition to a panel of binary interactions with known dissociation constants.

The first part of the manuscript describes and benchmarks the method, whereas the second part focuses on the interactions of one protein, CSP α , involved in synapse function.

General remarks

There are many well-established interaction assays that scientists can use, but the problem is that there is no single assay that works in every situation. In particular, transient interactions that are difficult to detect but biologically highly relevant have been challenging to study with many existing methods. Therefore, it is important that new assays are still developed and rigorously benchmarked.

Technically, the experiments described in the manuscript are well controlled and clearly presented, and the results are (mostly) convincing. I only have a few comments about the technical aspects of the manuscript (listed below).

As a method development paper, this manuscript is almost entirely technical in nature. The authors do not perform any follow-up experiments with the interactors detected in the CSP α screen, and therefore the biological significance of the interactions remains to be determined.

Our main intention with this study was to present the design, the benchmarking and the validation of a new, double-readout PPI detection method. To underline this point we have now changed the title of our study. The PPI screen with CSP α and the synaptic proteins was performed to provide proof-of-principle that the established method is capable of detecting relevant PPIs in a larger-scale interaction screening. Thus, a comprehensive validation of the biological significance of the newly detected CSP α interactions would go beyond the scope of this paper. Nevertheless, we now have validated some of the detected CSP α interactions in primary neurons using a proximity ligation assay (PLA), suggesting that also the endogenous proteins interact in cells. The experiments are described in the text (see **page 17, line 4-12**); the results are presented in **Appendix Fig. 14E**.

Major points

The number of interactors detected in the CSP α screen is, to me, alarmingly high (38% of tested pairs!). While it is possible that this is true (given that CSP α is a chaperone), it would be important to control this more rigorously. I suggest the authors test another membrane protein or membrane-associated protein for interactions with the same set to make sure these interactions are not due to crowding at the two-dimensional surface to which membrane proteins are tethered.

We agree with the referee that PPIs between membrane-associated proteins or proteins with membrane-spanning domains need to be carefully assessed and validated. We first reanalyzed the hPRS/hRRS reference sets (see also suggestion by Reviewer #1) for PPIs with membrane proteins and found that especially interactions where both tested proteins are membrane-associated or contain membrane domains are recovered with a lower specificity from the reference sets (see text **page 7, line 18 to page 8, line 2**; and **Appendix Fig. S5C-F**). We therefore defined more stringent cutoffs for the cBRET and cLuC ratios specifically for such interactions. Using the new cutoffs, we reanalyzed all PPIs with membrane proteins initially detected in the CSP α screen and performed a secondary validation screen with interactions above the cutoffs. This step did reduce the number of identified CSP α interaction partners with a membrane association (from initially 36 to now 22), suggesting that the remaining proteins indeed specifically associate with CSP α (**Appendix Tab. S6**).

Finally, as suggested by the reviewer, we performed an additional focused validation screen and investigated whether the membrane associated protein PICK1 also interacts with the 22 membrane proteins that interact with CSP α . The results are described in the text (see **page 16, line 21 to page 17, line 3** and presented in **Appendix Fig. S14B-D** and **Appendix Tab. S9**). Strikingly, we found no overlapping interactions with PICK1, supporting our hypothesis that CSP α interacts specifically with synaptic membrane proteins.

In the PRS/RRS experiment, were the positive hits from both orientations included in the analysis? In all previous assays (Y2H, MAPPIT, LUMIER etc), the interactions were tested in only one orientation, so comparison to the union of two different orientations in LuTHy is not appropriate. It would be better to show the performance of the assay with both orientations and compare them separately to the other established assays. In addition, I would suggest the authors add a supplementary figure where they plot the interaction scores from one orientation against the other to get an idea of the robustness of the assay.

While the PRS/RRS sets have indeed only been tested in one orientation initially using assays such as Y2H, MAPPIT or LUMIER (Braun, P. et al. (2009) An experimentally derived confidence score for binary protein-protein interactions. *Nat. Methods* 6, 91–97), the sets have been used consistently over the years with other assays such as KISS (Lievens, S. et al. (2014) Kinase Substrate Sensor (KISS), a mammalian in situ protein interaction sensor. *Mol Cell Proteomics* 13, 3332–3342) and DULIP (Trepte, P. et al. (2015) DULIP: A Dual Luminescence-Based Co-

Immunoprecipitation Assay for Interactome Mapping in Mammalian Cells. *Journal of Molecular Biology* 427, 3375–3388), where already two orientations were tested.

We reanalyzed the hPRS/hRRS reference sets for both orientations individually and included this analysis in **Appendix Tables S2 and S3**. For orientation 1, we find a sensitivity and specificity of 35.6% and 97.3%, respectively, while for orientation 2, a sensitivity of 43.8% and a specificity of 100% was determined. The interaction scores for both orientations of the hPRS and hRRS are displayed in **Appendix Figure 4C,D** and **Appendix Figure 5A,B**. All interaction scores for the two orientations can be found in **Appendix Tables S2 and S3**.

However, we do not believe that plotting the interactions scores of the two orientations against each other gives an indication of the robustness of the assay. The orientations tested have an impact on the obtained interaction scores (see new **Fig. 5**) resulting probably from factors like expression levels, proportion of donor/prey to acceptor/bait, affinity and competitive binding of endogenous proteins that can all influence the recovery of PPIs from reference sets.

To understand the impact of fusion tags and orientations more comprehensively, we have devised an independent experiment. We systematically tested the potential interactions between VCP and 10 UBX domain-containing proteins in all 8 possible combinations (PA-mCit and NL fusion tags at the N-terminus and the C-terminus). Importantly, we could validate all interactions between VCP and the 10 UBX domain-containing proteins, even though not all 8 orientations were positive for all interactions. For example, while all 8 orientations between VCP and UBXD9 were recovered, only 4 orientations were positive for the interaction between VCP and UBXD3 (please see **Fig. 5A-D** and **Appendix Tab. S5** for more details). To obtain the highest assays sensitivity, we can only advise to systematically test all possible 8 fusion tag orientations. The results are described in the text (see new text **page 13, line 21 to page 14, line 26**).

Because BRET is highly dependent on the proximity of the donor and acceptor, it would be nice to have an indication as to how much the interactions change when proteins are tagged in the C versus N terminus. "Where should I tag my favourite protein?" is a very common question for everyone working with fusion proteins, and such data would be very helpful to anyone who wants to use LuTHy assay in the future. I would encourage the authors to add such data if they have it.

To systematically address this important question the potential interactions between VCP and 10 selected UBX-domain containing proteins were investigated (see **Fig. 5A-D** and **Appendix Tab. S5**). We tested a total of 80 interactions between VCP and the 10 UBX-domain contacting proteins by analyzing all 8 possible orientations in BRET and LuC assays. We could indeed recover all previously described interactions (Raman, M. et al. (2015) Systematic proteomics of the VCP–UBXD adaptor network identifies a role for UBXN10 in regulating ciliogenesis. *Nat Cell Biol* 17, 1356–1369), but not in all orientations, indicating that the tag orientation indeed has an impact on the recovery rate of PPIs (**Appendix Tab. S5**).

Additionally, we analyzed the results more quantitatively to understand which orientation provides the highest BRET and LuC ratios. These results show nicely that for PPI detection with the BRET readout, the tags should be fused in close proximity to the interaction domain (here N-terminally on VCP and C-terminally on the UBX-domain containing proteins). In comparison, for LuC experiments it is more important to decide which of the two tested proteins of interest should be used as the PA-tagged bait. We e.g., found that PA-tagged VCP fusions work very efficiently as baits and recover a high number of PPIs from the interaction set (see new text **page 13, line 21 to page 14, line 26** and new **Figure 5D**).

The authors state that the CSP α preferentially targets membrane proteins, since 88% of interactors were of such kind. Please test if this is a significant enrichment over the starting pool, since synaptic proteins are likely enriched in membrane proteins to begin with.

This is an important point. We have performed a statistical test (see **Appendix Fig. S14A** and **Appendix Fig. S14H**) and observed that interactions of CSP α with membrane proteins were not significantly enriched over interactions with non-membrane proteins in the synaptic and non-synaptic interaction set. We have removed the statement that CSP α preferentially targets membrane proteins from the text.

It is not clear to me how the proteins screened for CSP α interactions were tagged (N terminus or C terminus). If they were tagged in the N terminus (as it seems), I would be very worried about proper trafficking of many membrane proteins since this would disrupt the signal sequence. Please comment.

The tag orientation was chosen for membrane proteins according to their “architecture” in the membrane, so that the fusion tag would be located in the cytoplasm. Similarly, if the protein contained a membrane anchor, the fusion tag was added to the protein on the opposite side in order not to disrupt membrane localization. All tag orientations are indicated in **Appendix Tables S6 and S10**.

Minor points

The authors state (page 14) that LuC is less suitable for detecting changes in interactions due to mutations. However, they only tested with LuC those interactions that were detected as changed by BRET. Thus, it is possible that 1) LuC is less suitable (as they claim), 2) BRET is less suitable because the changes it detected are not real, or 3) LuC is more suitable because it would have detected other changed interactions that BRET did not detect, or 4) they are equally suitable.

We have tested all wild-type CSP α interactions that were initially detected with the BRET (36 PPIs) or LuC (7 PPIs) readout against the two disease mutants (see **Appendix Tab. S11**). We detected with the BRET readout that 21 (Δ L116) and 19 (L115R) of the 36 CSP α wild-type PPIs were affected by the mutations, respectively. With LuC, we found that 2 (Δ L116) and 3 (L115R) of the 7 CSP α wild-type PPIs were affected. This suggests that both readouts are suitable to detect the effects of disease-causing mutations on interactions. The previous statement that LuC is less suitable for detecting changes in interactions was removed from the text.

The authors state that BRET detects direct interactions, because the donor and the acceptor have to be in close proximity. This is incorrect, since they can be in close proximity even if the interaction is not direct.

We agree that interactions can be bridged by a third protein also in BRET experiments, without being in direct contact. However, it seems to us that this may be a very unlikely event. We believe that in comparison to co-precipitation-based assays (Zhang, X.-F., Ou-Yang, L., Hu, X. & Dai, D.-Q. (2015) Identifying binary protein-protein interactions from affinity purification mass spectrometry data. *BMC Genomics* 16, 745 and Las Rivas, De, J. & Fontanillo, C. (2010) Protein-protein interactions essentials: key concepts to building and analyzing interactome networks. *PLoS Comput Biol* 6, e1000807), in BRET, FRET or protein complementation assays such as BiFC the likelihood is very high that proteins of interest directly interact. We therefore feel that our cautious statement that “positive BRET indicates direct binding” can remain in the text (page **21**, lines **7-8**).

On page 6 (and in several graphs), the authors mention measuring luminescence. Technically, the emitted light is not only luminescence but a combination of luminescence (photons emitted by Nanoluc) and fluorescence (photons emitted by mCitrine).

We agree that the detected light is a combination of photons emitted from the luciferase and the fluorescent protein. However, we believe that we have used the term luminescence correctly. The term was first introduced in 1888 by the German physicist Eilhardt Wiedemann. He described luminescence for “all those phenomena of light which are not solely conditioned by the rise in temperature” (for example, glowing of metal when heated; from “A history of Luminescence”, 1957, E. Newton Harvey, <https://archive.org/stream/historyoflumines00harv#page/n9/mode/2up>). Therefore, the measured light is luminescence, as it is technically a combination of bioluminescence (photons emitted by the luciferase due to the oxidation of the substrate) and of fluorescence (photons emitted by mCitrine through the energy transfer from the luciferase).

Page 10: co-produced -> co-expressed

The wording was changed throughout the text.

Reviewer #3:

The LuTHy assay described by Trepte and colleagues combines two orthogonal protein-protein interaction (PPI) assays, Bioluminescence Resonance Energy Transfer (BRET) and luminescence-based co-precipitation (LuC), as a new improved method for the mapping of PPIs. This clever approach deals with a typical validation strategy for PPIs detected with a particular assay, i.e. the confirmation in an orthogonal assay. The authors rightfully point out that the high heterogeneity underlying PPIs requires complementary approaches and they refer to important papers in the field. The authors indeed suggest to combine the data from the two assays as a robust sensitive research tool to obtain a comprehensive view on the interactome.

The different aspects related to state-of-the-art binary interactomics are nicely addressed in this study: small molecule effects, weak interactions, donor saturation assays for binding constants, and interactome profiling of pathogenic variants.

The manuscript is clearly written and well structured. Figures are clear and to the point. Sufficient additional figures are provided to assess validity of all the claims. Overall quality of the work is very high.

We want to thank the reviewer for her/his very positive statement.

While experimentally sound, and a nice contribution to the binary interactomics field, one important aspect is not handled in this manuscript. The plasmids used in this study contain CMV promoters. Combined with PEI-based transfections in HEK293 cells, it can be assumed that extremely high levels of both bait and prey are obtained in the cells. This can have different effects on the cells themselves (various levels of overexpression stress) and can also lead to artifacts such as false positives. While this is controlled to some extent by the random reference set (and the non-synapse set), this point should be addressed in the manuscript. Although the donor saturation assay uses increasing levels of one of the partners, it is not the best way to show this. Ideally, the authors should show the system for lower expression levels. Lentiviral transduction or stable integration of bait and prey can be considered, preferably in a different cellular background (for instance: in the SHEP cells).

We agree with the reviewer that protein expression levels should preferentially be low when PPIs between tagged proteins are assessed in cell models. To address this key point we have performed two additional experiments:

- 1) We transfected different amounts of plasmid DNAs into HEK293 cells and compared the transfected fusion protein levels to their respective endogenous protein levels by immunoblotting (see text **page 15, line 1-16** and **Appendix Fig. S11A-E**). Importantly, luminescence emission could still be observed when transfecting only 1 ng of plasmid DNA per well of a 96-well plate. Under these conditions, the levels of the fusion-protein NL-VCP were significantly lower than endogenous VCP. Similarly, fluorescence

could still be measured when transfecting as little as 25 ng of the fluorescent protein UBXD9-mCit, which showed comparable levels to the endogenous protein. When co-transfecting 1 ng of NL-VCP and 25 ng of UBXD9-mCit, a significant BRET ratio was determined, compared to the two tested control interactions NL-VCP/PA-mCit and NL/UBXD9-mCit. This shows that the LuTHy assay is very sensitive and well suited to detect PPIs at very low protein concentrations.

- 2) We also generated new lentiviral expression vectors that contain the relatively weak UBC promoter. We generated viral particles and infected primary neurons for the expression of CSP α -mCit-PA and NL-ZDHHC17. Strikingly, the interaction between CSP α and ZDHHC17 was successfully validated with both BRET and LuC readouts. Furthermore, we confirmed the effects of the disease-causing mutations L116 Δ and L115R on this interaction. The new results are shown in **Fig. 6E** and described on **page 19 line 22 to page 20, line 2** in the text.

I have mentioned higher that an orthogonal approach is a good validation strategy. Actually, most studies now indeed aim at orthogonal validation, but at the endogenous level. As an alternative, the authors can consider to evaluate some of the found interactions for CSP α at the endogenous level. It is clear however that forced expression is a typical drawback with most binary approaches.

We agree with the referee and have investigated whether selected CSP α interactions can be validated with proximity ligation assays (PLAs) in neuronal cells. The PLA method enables the detection of protein associations with the help of protein-specific antibodies in cells at endogenous levels. We successfully validated all 5 selected CSP α interactions that were initially identified with the LuTHy method (see **Appendix Fig. S14E**). In contrast, an interaction between CSP α and STX4 was not detected with PLA. Initially, an interaction between these two proteins was also not observed with the LuTHy method (negative control). Thus, PPIs that are identified with LuTHy can be successfully validated at the endogenous protein levels with orthogonal assays. The experiments are described in the text (see **page 17, line 4-12**).

Minor points:

Please explain how the selection was made to obtain the panel of interactions for the affinity range. What criteria (if any) were used?

We used the PDBbind and PPI affinity database to assemble an initial set of protein-protein interactions with known binding affinities. From those, we tested the interactions for which we had cDNA clones in the lab.

How do the cut-offs defined in the materials and methods (p26 first paragraph; ≥ 0.03 and ≥ 0.05) relate to the data in the table EV6?

We have now changed the analysis of the data in **Appendix Table S6**. Since we did not include all controls NL/ CSP α -mCit-PA and NL-X/PA-mCit in our initial primary screen of the CSP α -mCit-PA protein against the 125 NL-tagged synaptic proteins, we were only able to calculate BRET and LuC ratios, but not corrected BRET and LuC ratios (cBRET and cLuC, see **Appendix Figure S1F-G**). We therefore initially used different cut-offs, which we now changed because we rescreened all positive interactions against the respective controls. This allowed us to calculate cBRET and cLuC ratios for which we used the same cut-offs that were determined with hPRS and hRRS. See **Appendix Table S6** – secondary validation.

Figure 4A, Figure 5D: Is the data normalized for expression levels?

The data in Figures 4A (now **Appendix Figure S10B**) is not normalized for expression levels, but no significant changes in expression with the mutant proteins were observed (see **Appendix Figure S10A**). Figure 5D was removed from the manuscript.

Please also address what kind of false positives can be expected for the individual approaches, and for the combined assay (if any)

We believe that both approaches, BRET and LuC, could result in false-positives if high protein expression levels are used, due to crowding effects. We therefore routinely perform the transfection experiments with low DNA concentrations. Also, interactions that are only detected with the LuC readout, might be bridged by a third protein and the detected interactions might not be binary. Please, also see **page 21, lines 12-18** in the text.

2nd Editorial Decision

6th June 2018

Thank you for sending us your revised manuscript. We have now heard back from reviewer #2 who was asked to evaluate your revised study. As you will see below, the reviewer is satisfied with the modifications made and thinks that the study is now suitable for publication.

Before we formally accept the study for publication we would ask you to address a few minor editorial issues listed below.

Reviewer #2:

The authors have very convincingly addressed all the points I raised in the first review. I congratulate the authors on a very thorough investigation and the novel assay that will be useful to many people in the proteomics field!

Corresponding Author Name: Erich Wanker
 Journal Submitted to: Molecular Systems Biology
 Manuscript Number: MSB-17-8071RRR